# Cardiovascular outcomes of semaglutide and tirzepatide for patients with type 2 diabetes in clinical practice

Nils Krüger [1,2,3,4] ✉, Sebastian Schneeweiss[1,2], Rishi J. Desai [1,2], Sushama Kattinakere Sreedhara [1], Anna R. Kehoe[1,2], Kenshiro Fuse[1,2], Georg Hahn[1,2], Heribert Schunkert [3,4] & Shirley V. Wang [1,2]

Cardiovascular outcome trials of the incretin-based medicines tirzepatide and semaglutide have shown benefits in populations with varying levels of cardiovascular risk. However, without direct head-to-head comparisons, treatment decisions rely on indirect evidence from heterogeneous trial populations, leaving optimal treatment choices uncertain. Here we conducted five cohort studies to assess the effectiveness of tirzepatide and semaglutide in patients with elevated cardiovascular risk, including obesity and type 2 diabetes, enrolled in insurance programs in the USA between 2018 and 2025. First, we emulated two cardiovascular outcome trials, SUSTAIN-6 (semaglutide versus sitagliptin as placebo proxy) and SURPASS-CVOT (tirzepatide versus dulaglutide), to benchmark and critically evaluate our design, data and analytic framework. Second, we assessed each drug in expanded populations reflective of patients routinely seen in clinical practice. Third, we directly compared tirzepatide versus semaglutide. Baseline confounders were balanced using propensity score matching. For the primary composite endpoint of myocardial infarction, stroke or all-cause mortality, benchmarking identified high agreement between the reference trials and their emulations for all individual endpoints except for all-cause mortality in SUSTAIN-6, informing subsequent analyses. In expanded populations, comparing semaglutide versus sitagliptin for the composite outcome of myocardial infarction or stroke yielded a hazard ratio of 0.82 (95% confidence interval (CI) 0.74 to 0.91), and comparing tirzepatide versus dulaglutide for the composite outcome including mortality yielded a hazard ratio of 0.87 (95% CI 0.75 to 1.01). In the head-to-head comparison of tirzepatide versus semaglutide, the hazard ratio was 1.06 (95% CI 0.95 to 1.18). These findings support a comparable cardiovascular benefit of tirzepatide and semaglutide in clinical practice and demonstrate how rigorously designed real-world evidence can complement randomized clinical trials. ClinicalTrials.gov registration: NCT06659744, NCT07088718, NCT07096063.

Cardiovascular disease remains the leading cause of mortality worldwide, with obesity and type 2 diabetes as major modifiable risk factors[1]. In the USA, more than 40% of adults are obese, with projections indicating that one in two people will be affected by 2030 (refs. [2–4]). When present alongside other cardiovascular risk factors, obesity and type 2 diabetes compound the risk of myocardial infarction, stroke and premature death, escalating both individual and public health burden[5,6].

Glucagon-like peptide-1 (GLP-1) receptor agonists have emerged as key therapies for the treatment of obesity and type 2 diabetes, with several agents demonstrating cardiovascular benefits[7–12]. Among these, semaglutide has demonstrated reduced risk of major adverse cardiovascular events (MACE) in trial participants with moderate or high cardiovascular risk. Tirzepatide, a dual glucose-dependent insulinotropic polypeptide (GIP) and GLP-1 receptor agonist with rapidly growing use, has demonstrated even greater effects on glycemic control and weight loss[13–15]. However, evidence on the cardiovascular benefits of tirzepatide is only emerging and no randomized study has directly compared it to semaglutide[16,17]. In the absence of head-to-head comparisons, indirect evidence across heterogeneous trial populations with placebo or inferior active comparators provides limited guidance for clinical decision-making.

To provide timely, complementary evidence, we conducted a comparison of tirzepatide and semaglutide in patients at low, moderate and high cardiovascular risk who were diagnosed with obesity and type 2 diabetes and subgroups with specific cardiovascular conditions. Our study proceeded in three steps. First, we emulated the two cardiovascular outcome trials SUSTAIN-6 (semaglutide versus placebo) and SURPASS-CVOT (tirzepatide versus dulaglutide) using the RCT-DUPLICATE approach to benchmark findings and inform the study design and analytic approach for expanded questions; second, we evaluated the effectiveness of each agent in populations expanded to those treated in routine care; and third, we directly compared tirzepatide and semaglutide in contemporary patient populations reflective of clinical practice[18–21].

## Results

### Emulating pivotal cardiovascular outcome trials and expanding populations

To validate our analytical framework, we emulated the cardiovascular outcome trials SUSTAIN-6 and SURPASS-CVOT using three national claims databases from the USA. Protocol components, including eligibility criteria, treatment strategies and follow-up definitions, were closely aligned with the original trials (Methods). At the time the protocols were finalized, trial results for SURPASS-CVOT were not yet available; findings were released during the conduct of this study[16]. A total of 158,310 patients met the eligibility criteria for the emulation of SUSTAIN-6 and 44,671 patients for the emulation of SURPASS-CVOT.

When expanding trial eligibility criteria to reflect broader patient populations typically encountered in clinical practice, we identified 453,201 individuals initiating semaglutide or sitagliptin (expanding SUSTAIN-6 eligibility) and 136,089 initiating tirzepatide or dulaglutide (expanding SURPASS-CVOT eligibility). For the head-to-head comparison of tirzepatide versus semaglutide, 297,842 initiators were included (Fig. 1).

Before matching, patients in the trial eligible and expanded populations initiating semaglutide or tirzepatide were younger, more likely to be white and were more frequently prescribed sulfonylureas compared to sitagliptin or dulaglutide users. After 1:1 propensity score matching, measured baseline characteristics were well-balanced across treatment groups. In the matched expanded populations, the mean age ranged from 59.2 to 69.2 years, 50.4–55.8% were female and the mean body mass index (BMI) ranged from 34.5 to 38.7 kg m$^{-2}$. A history of prior myocardial infarction or stroke was present in 2.9–9.3% and chronic kidney disease (CKD) was observed in 18.9–36.7% of patients (Table 1 and Supplementary Tables 1–18).

### Benchmarking against SUSTAIN-6 and SURPASS-CVOT

In the emulation of SUSTAIN-6 comparing semaglutide to sitagliptin, a proxy for placebo, in patients at moderate and high cardiovascular risk (Table 2 and Supplementary Table 19), the hazard ratio (HR) for the primary endpoint was 0.68 (95% confidence interval (CI) 0.60–0.77) compared to the trial estimate of HR 0.74 (95% CI 0.58 to 0.95). The four agreement metrics were met (Table 2). When examining the individual components of the primary endpoint, we observed closely concordant results for myocardial infarction and stroke but divergent results for all-cause mortality, suggestive of residual confounding. Secondary endpoints in the trial emulations as well as in their respective reference trials were not powered sufficiently to assess statistical agreement.

In the emulation of SURPASS-CVOT comparing tirzepatide to dulaglutide in patients at high cardiovascular risk (Table 2 and Supplementary Table 19), the estimated HR was 0.83 (95% CI 0.69 to 1.01) compared to the trial estimates of HR 0.92 (95% CI 0.83 to 1.01). This benchmarking confirmed all agreement metrics, for both the primary composite outcome and all-cause mortality, supporting the validity of our approach.

### Applying learnings from benchmarks to expanded populations

Informed by the database study that benchmarked against SUSTAIN-6, we amended the protocol of the expansion study comparing semaglutide versus sitagliptin to focus on endpoints that did not include death of any cause. Specifically, we added a composite endpoint of myocardial infarction or stroke without death of any cause. Similarly, a version of the composite endpoint of hospitalization for heart failure or urgent care visit requiring intravenous diuretics was added that did not include death. Amendments were documented in updated study protocols available on ClinicalTrials.gov. No changes were made to the endpoints for the comparison of tirzepatide versus dulaglutide in the expanded population because we observed high concordance with SURPASS-CVOT estimates in the primary composite endpoint as well as mortality in the benchmarking study. As the confounding structure for the head-to-head comparison of tirzepatide versus semaglutide was expected to be more similar to the tirzepatide versus dulaglutide benchmarking study, we proceeded with the prespecified analysis plan that included death in the primary composite MACE outcome.

### Primary endpoint in expanded populations and high-risk subgroups

**Semaglutide versus sitagliptin.** Among patients with obesity and type 2 diabetes at low, moderate or high cardiovascular risk in clinical practice, the 1-year risk of the composite endpoint of myocardial infarction or stroke was 1.5% (95% CI 1.4% to 1.6%) with semaglutide compared to 1.7% (95% CI 1.6% to 1.9%) with sitagliptin. This corresponded to a risk difference of −0.3% (95% CI −0.4% to −0.1%) and a hazard ratio of 0.82 (95% CI 0.74 to 0.91) (Fig. 2 and Supplementary Table 19). Pooled mean follow-up on-treatment for semaglutide users was 193 days (median 157 days, interquartile range (IQR) 85 to 331 days) and for sitagliptin users 195 days (median 160 days, IQR 95 to 322 days). Treatment discontinuation (46%) was the most common reason for censoring (Supplementary Table 20). In the subgroup at high cardiovascular risk, effect estimates were similar (HR 0.80, 95% CI 0.71 to 0.91).

**Tirzepatide versus dulaglutide.** Among patients in the expanded population who initiated tirzepatide or dulaglutide, the 1-year risk for the primary endpoint including all-cause mortality in the tirzepatide group was 1.4% (95% CI 1.3% to 1.6%) versus 1.8% (95% CI 1.5% to 2.0%) for dulaglutide. This yielded a risk difference of −0.3% (95% CI −0.6% to 0.04%) and an HR of 0.87 (95% CI 0.75 to 1.01) (Fig. 2 and Supplementary Table 19). Among tirzepatide users, the pooled mean

**a**

Benchmark against and predict trial results

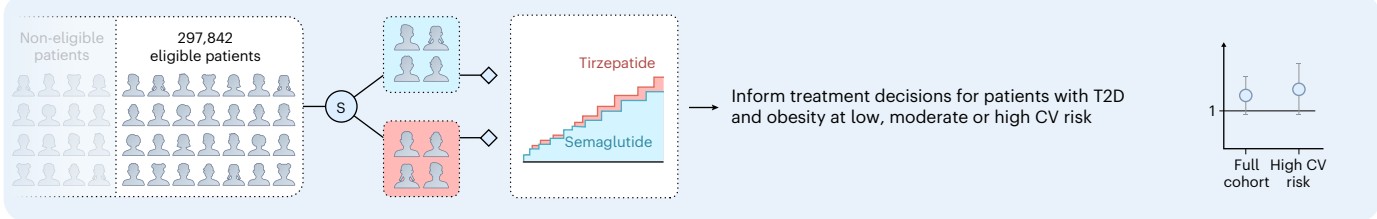

**b** Study effects in expanded populations treated in clinical practice and explore subgroups of clinical interest

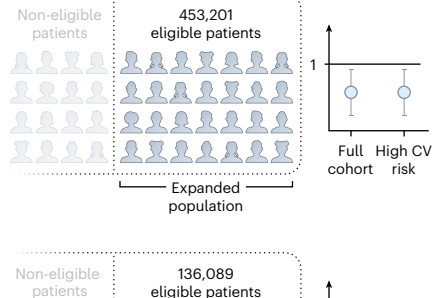

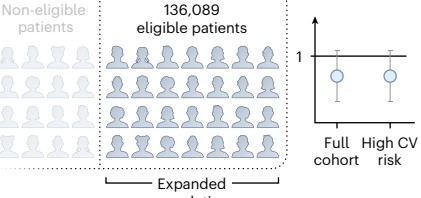

**c** Assess the comparative effects of tirzepatide against semaglutide in patients routinely seen in clinical practice

**Fig. 1 | Overview of the study design to assess the comparative effects of semaglutide and tirzepatide in patients at cardiovascular risk. a–c**, The study proceeded in three sequential steps: (1) we emulated the design of the SUSTAIN-6 and SURPASS-CVOT trials using three US claims databases to benchmark the trial emulations against the the reference trials and predict results (**a**); (2) we expanded the patient populations within this framework to assess the effectiveness of each agent in clinical practice (**b**); and (3) we compared tirzepatide versus semaglutide in a head-to-head comparison to inform clinical decision-making (**c**). Error bars represent the 95% CI of the point estimates. CV, cardiovascular; S, selection of patients initiating semaglutide, tirzepatide, dulaglutide or sitagliptin (as a placebo proxy) via propensity score 1:1 nearest neighbor matching to mimic randomization; T2D, type 2 diabetes.

on-treatment follow-up was 189 days (median 162 days, IQR 72 to 321 days). For dulaglutide users, the corresponding follow-up was, on average, 173 days (median 139 days, IQR 73 to 257 days). Discontinuation of treatment (37%) was the most common reason for censoring (Supplementary Table 20). Among patients at high cardiovascular risk, the effect estimate was an HR of 0.88 (95% CI 0.73 to 1.07).

**Tirzepatide versus semaglutide.** In the direct head-to-head comparison, the 1-year risk of the primary endpoint including all-cause mortality was 1.3% (95% CI 1.2% to 1.5%) for tirzepatide and 1.3% (95% CI 1.2% to 1.5%) for semaglutide, resulting in a risk difference of 0.0% (95% CI −0.2% to 0.2%) and an HR of 1.06 (95% CI 0.95 to 1.18) (Fig. 2 and Supplementary Table 19). The mean follow-up on-treatment was 181 days (median 155 days, IQR 71 to 290 days) for tirzepatide initiators and 174 days (median 148 days, IQR 82 to 254 days) for semaglutide initiators. Treatment discontinuation (34%) was the most common censoring reason (Supplementary Table 20). Among individuals at high cardiovascular risk, the HR was 1.11 (95% CI 0.96 to 1.27).

**Secondary, safety and negative control endpoints**
For individual components of the primary endpoint, semaglutide versus sitagliptin reduced myocardial infarction (HR 0.81, 95% CI 0.70 to 0.92) and stroke (HR 0.84, 95% CI 0.71 to 0.99) in the expanded populations at low, moderate and high cardiovascular risk. Tirzepatide showed a noninferior reduction in all-cause mortality (HR 0.88, 95% CI 0.68 to 1.16), myocardial infarction (HR 0.91, 95% CI 0.73 to

1.12) and stroke (HR 0.78, 95% CI 0.59 to 1.03) compared to dulaglutide, although CIs remained compatible with no difference. In the head-to-head comparison between tirzepatide and semaglutide, the two drugs yielded similar risks for component endpoints all-cause mortality (HR 1.03, 95% CI 0.84 to 1.27), myocardial infarction (HR 1.03, 95% CI 0.88 to 1.21) and stroke (HR 1.15, 95% CI 0.92 to 1.45) (Fig. 3 and Supplementary Table 21).

Semaglutide showed a lower risk of the secondary composite endpoint of heart failure hospitalization or urgent heart failure visit compared to sitagliptin in the expanded populations (HR 0.61, 95% CI 0.57 to 0.66). Tirzepatide demonstrated a lower risk of the secondary composite endpoint in heart failure hospitalization, urgent heart failure visit or all-cause mortality compared to dulaglutide HR 0.75 (95% CI 0.65 to 0.86). For tirzepatide versus semaglutide, contrary to the primary endpoint in the head-to-head comparison, there was some supporting evidence for tirzepatide to have a modest benefit, although the 95% CI overlapped the null (HR 0.91, 95% CI 0.81 to 1.01) (Fig. 3 and Supplementary Table 21).

For safety outcomes, semaglutide and tirzepatide showed lower risks for serious bacterial infections compared to sitagliptin and dulaglutide, respectively. No meaningful differences in the risk of urinary tract infections or gastrointestinal adverse events between treatment groups were observed (Supplementary Table 22).

No associations were observed for the two negative control outcomes, lumbar radiculopathy and abdominal hernia, supporting the validity of the analyses (Fig. 3 and Supplementary Table 23).

**Table 1 | Baseline characteristics of patients initiating semaglutide versus sitagliptin, tirzepatide versus dulaglutide and tirzepatide versus semaglutide after 1:1 propensity score matching**

| | n (%) | | | | | | | | |
|---|---|---|---|---|---|---|---|---|---|
| | **Semaglutide versus sitagliptin** | | | **Tirzepatide versus dulaglutide** | | | **Tirzepatide versus semaglutide** | | |
| | Semaglutide (n = 79,501) | Sitagliptin (n = 79,501) | SMD | Tirzepatide (n = 39,152) | Dulaglutide (n = 39,152) | SMD | Tirzepatide (n = 86,191) | Semaglutide (n = 86,191) | SMD |
| Demographics | | | | | | | | | |
| Age (years); mean (s.d.) | 63.28 (11.18) | 63.32 (11.89) | 0.00 | 60.38 (11.45) | 60.38 (11.72) | 0.00 | 59.24 (11.59) | 59.26 (11.72) | 0.00 |
| Gender | | | | | | | | | |
| Female; n (%) | 42,525 (53.5%) | 42,657 (53.7%) | 0.00 | 21,157 (54.0%) | 21,181 (54.1%) | 0.00 | 48,025 (55.7%) | 48,074 (55.8%) | 0.00 |
| Male; n (%) | 36,976 (46.5%) | 36,844 (46.3%) | 0.00 | 17,995 (46.0%) | 17,971 (45.9%) | 0.00 | 38,166 (44.3%) | 38,117 (44.2%) | 0.00 |
| Race | | | | | | | | | |
| White; n (%) | 25,746 (55.6%) | 25,740 (55.6%) | 0.00 | 10,954 (28.0%) | 10,940 (27.9%) | 0.00 | 27,648 (32.1%) | 27,589 (32.0%) | 0.00 |
| Black; n (%) | 6,321 (13.6%) | 6,254 (13.5%) | 0.00 | 3,483 (8.9%) | 3,471 (8.9%) | 0.00 | 7,551 (8.8%) | 7,535 (8.7%) | 0.00 |
| Unknown/missing; n (%) | 13,038 (28.2%) | 13,114 (28.3%) | 0.00 | 9,991 (25.5%) | 10,009 (25.6%) | 0.00 | 24,779 (28.7%) | 24,866 (28.8%) | 0.00 |
| Others; n (%) | 1,207 (2.6%) | 1,204 (2.6%) | 0.00 | 367 (0.9%) | 375 (1.0%) | 0.00 | 868 (1.0%) | 856 (1.0%) | 0.00 |
| Region/state | | | | | | | | | |
| Northeast; n (%) | 10,455 (13.2%) | 10,520 (13.2%) | 0.00 | 3,689 (9.4%) | 3,703 (9.5%) | 0.00 | 6,789 (7.9%) | 6,688 (7.8%) | 0.00 |
| Midwest/North Central; n (%) | 15,841 (19.9%) | 15,989 (20.1%) | 0.01 | 9,574 (24.5%) | 9,609 (24.5%) | 0.00 | 18,664 (21.7%) | 18,693 (21.7%) | 0.00 |
| South; n (%) | 43,242 (54.4%) | 43,187 (54.3%) | 0.00 | 20,441 (52.2%) | 20,342 (52.0%) | 0.01 | 50,849 (59.0%) | 50,808 (58.9%) | 0.00 |
| West; n (%) | 9,901 (12.5%) | 9,742 (12.3%) | 0.01 | 5,430 (13.9%) | 5,477 (14.0%) | 0.00 | 9,831 (11.4%) | 9,944 (11.5%) | 0.00 |
| BMI class (kg m⁻²) | | | | | | | | | |
| 25.0–29.9; n (%) | 13,016 (16.4%) | 12,821 (16.1%) | 0.01 | 3,601 (9.2%) | 3,610 (9.2%) | 0.00 | 5,712 (6.6%) | 5,723 (6.6%) | 0.00 |
| 30.0–34.9; n (%) | 9,831 (12.4%) | 9,751 (12.3%) | 0.00 | 6,388 (16.3%) | 6,401 (16.3%) | 0.00 | 12,744 (14.8%) | 12,713 (14.7%) | 0.00 |
| 35.0–39.9; n (%) | 5,791 (7.3%) | 5,829 (7.3%) | 0.00 | 8,502 (21.7%) | 8,501 (21.7%) | 0.00 | 20,622 (23.9%) | 20,589 (23.9%) | 0.00 |
| 40.0 and above; n (%) | 22,425 (28.2%) | 22,492 (28.3%) | 0.00 | 9,925 (25.3%) | 9,921 (25.3%) | 0.00 | 24,374 (28.3%) | 24,478 (28.4%) | 0.00 |
| Unspecified obesity; n (%) | 28,438 (35.7%) | 28,608 (36.0%) | 0.00 | 10,736 (27.4%) | 10,719 (27.4%) | 0.00 | 22,739 (26.4%) | 22,688 (26.3%) | 0.00 |
| Cardiovascular risk factors | | | | | | | | | |
| Smoking/tobacco use; n (%) | 15,188 (19.1%) | 15,201 (19.1%) | 0.00 | 6,039 (15.4%) | 6,003 (15.3%) | 0.00 | 16,213 (18.8%) | 16,255 (18.9%) | 0.00 |
| Hypertension; n (%) | 68,720 (86.4%) | 68,763 (86.5%) | 0.00 | 33,542 (85.7%) | 33,459 (85.5%) | 0.01 | 73,529 (85.3%) | 73,475 (85.2%) | 0.00 |
| Hyperlipidemia; n (%) | 65,752 (82.7%) | 65,555 (82.5%) | 0.01 | 32,542 (83.1%) | 32,485 (83.0%) | 0.00 | 71,467 (82.9%) | 71,458 (82.9%) | 0.00 |
| Cardiovascular comorbidities | | | | | | | | | |
| Coronary atherosclerosis; n (%) | 16,317 (20.5%) | 16,130 (20.3%) | 0.01 | 7,030 (18.0%) | 6,973 (17.8%) | 0.00 | 14,415 (16.7%) | 14,483 (16.8%) | 0.00 |
| Stable angina; n (%) | 3,301 (4.2%) | 3,251 (4.1%) | 0.00 | 1,464 (3.7%) | 1,472 (3.8%) | 0.00 | 3,052 (3.5%) | 3,071 (3.6%) | 0.00 |
| Unstable angina; n (%) | 2,046 (2.6%) | 2,033 (2.6%) | 0.00 | 809 (2.1%) | 796 (2.0%) | 0.00 | 1,610 (1.9%) | 1,617 (1.9%) | 0.00 |
| Acute myocardial infarction; n (%) | 1,008 (1.3%) | 990 (1.2%) | 0.00 | 390 (1.0%) | 396 (1.0%) | 0.00 | 671 (0.8%) | 692 (0.8%) | 0.00 |
| Old myocardial infarction; n (%) | 3,238 (4.1%) | 3,192 (4.0%) | 0.00 | 1,296 (3.3%) | 1,288 (3.3%) | 0.00 | 2,513 (2.9%) | 2,506 (2.9%) | 0.00 |
| Cardiac conduction disorder; n (%) | 4,154 (5.2%) | 4,115 (5.2%) | 0.00 | 1,862 (4.8%) | 1,863 (4.8%) | 0.00 | 3,779 (4.4%) | 3,772 (4.4%) | 0.00 |
| Previous cardiac procedure; n (%) | 1,537 (1.9%) | 1,467 (1.8%) | 0.01 | 555 (1.4%) | 544 (1.4%) | 0.00 | 1,029 (1.2%) | 1,065 (1.2%) | 0.00 |
| Ischemic stroke; n (%) | 821 (1.0%) | 822 (1.0%) | 0.00 | 263 (0.7%) | 274 (0.7%) | 0.00 | 408 (0.5%) | 400 (0.5%) | 0.00 |
| TIA; n (%) | 1,331 (1.7%) | 1,349 (1.7%) | 0.00 | 572 (1.5%) | 577 (1.5%) | 0.00 | 1,109 (1.3%) | 1,132 (1.3%) | 0.00 |
| Peripheral vascular disease or surgery; n (%) | 7,128 (9.0%) | 7,027 (8.8%) | 0.00 | 3,097 (7.9%) | 3,099 (7.9%) | 0.00 | 5,935 (6.9%) | 5,900 (6.8%) | 0.00 |
| Atrial fibrillation; n (%) | 7,657 (9.6%) | 7,722 (9.7%) | 0.00 | 3,167 (8.1%) | 3,160 (8.1%) | 0.00 | 6,667 (7.7%) | 6,755 (7.8%) | 0.00 |
| Other cardiac dysrhythmia; n (%) | 14,397 (18.1%) | 14,437 (18.2%) | 0.00 | 6,857 (17.5%) | 6,842 (17.5%) | 0.00 | 15,194 (17.6%) | 15,379 (17.8%) | 0.01 |
| Edema; n (%) | 9,313 (11.7%) | 9,315 (11.7%) | 0.00 | 4,542 (11.6%) | 4,452 (11.4%) | 0.01 | 10,018 (11.6%) | 10,029 (11.6%) | 0.00 |
| Heart failure; n (%) | 9,656 (12.1%) | 9,516 (12.0%) | 0.01 | 4,405 (11.3%) | 4,289 (11.0%) | 0.01 | 8,641 (10.0%) | 8,761 (10.2%) | 0.01 |
| Cardiomyopathy; n (%) | 3,661 (4.6%) | 3,619 (4.6%) | 0.00 | 1,674 (4.3%) | 1,671 (4.3%) | 0.00 | 3,290 (3.8%) | 3,346 (3.9%) | 0.00 |
| Valve disorders; n (%) | 7,803 (9.8%) | 7,812 (9.8%) | 0.00 | 3,444 (8.8%) | 3,444 (8.8%) | 0.00 | 7,338 (8.5%) | 7,343 (8.5%) | 0.00 |
| Valve replacement; n (%) | 915 (1.2%) | 925 (1.2%) | 0.00 | 341 (0.9%) | 347 (0.9%) | 0.00 | 690 (0.8%) | 675 (0.8%) | 0.00 |
| Implantable cardioverter defibrillator; n (%) | 234 (0.3%) | 225 (0.3%) | 0.00 | 100 (0.3%) | 94 (0.2%) | 0.00 | 184 (0.2%) | 203 (0.2%) | 0.01 |
| Pulmonary hypertension; n (%) | 1,977 (2.5%) | 2,009 (2.5%) | 0.00 | 911 (2.3%) | 906 (2.3%) | 0.00 | 1,972 (2.3%) | 1,962 (2.3%) | 0.00 |
| Venous or pulmonary embolism; n (%) | 2,405 (3.0%) | 2,434 (3.1%) | 0.00 | 1,159 (3.0%) | 1,143 (2.9%) | 0.00 | 2,417 (2.8%) | 2,411 (2.8%) | 0.00 |

**Table 1 (continued) | Baseline characteristics of patients initiating semaglutide versus sitagliptin, tirzepatide versus dulaglutide and tirzepatide versus semaglutide after 1:1 propensity score matching**

| | n (%) | | | | | | | | |
|---|---|---|---|---|---|---|---|---|---|
| | Semaglutide versus sitagliptin | | | Tirzepatide versus dulaglutide | | | Tirzepatide versus semaglutide | | |
| | Semaglutide (n = 79,501) | Sitagliptin (n = 79,501) | SMD | Tirzepatide (n = 39,152) | Dulaglutide (n = 39,152) | SMD | Tirzepatide (n = 86,191) | Semaglutide (n = 86,191) | SMD |
| Diabetes complications | | | | | | | | | |
| Diabetic retinopathy; n (%) | 7,018 (8.8%) | 6,895 (8.7%) | 0.01 | 3,520 (9.0%) | 3,540 (9.0%) | 0.00 | 5,979 (6.9%) | 5,944 (6.9%) | 0.00 |
| Diabetic neuropathy; n (%) | 20,103 (25.3%) | 19,893 (25.0%) | 0.01 | 9,816 (25.1%) | 9,731 (24.9%) | 0.01 | 17,753 (20.6%) | 17,886 (20.8%) | 0.00 |
| Diabetic nephropathy; n (%) | 16,382 (20.6%) | 16,090 (20.2%) | 0.01 | 7,859 (20.1%) | 7,673 (19.6%) | 0.01 | 14,167 (16.4%) | 14,227 (16.5%) | 0.00 |
| Diabetes with peripheral circulatory disorders; n (%) | 9,605 (12.1%) | 9,583 (12.1%) | 0.00 | 5,044 (12.9%) | 4,988 (12.7%) | 0.00 | 9,627 (11.2%) | 9,683 (11.2%) | 0.00 |
| Diabetic foot; n (%) | 2,096 (2.6%) | 2,073 (2.6%) | 0.00 | 1,106 (2.8%) | 1,077 (2.8%) | 0.00 | 1,978 (2.3%) | 1,957 (2.3%) | 0.00 |
| Erectile dysfunction; n (%) | 3,153 (4.0%) | 3,084 (3.9%) | 0.00 | 1,811 (4.6%) | 1,827 (4.7%) | 0.00 | 3,900 (4.5%) | 3,988 (4.6%) | 0.01 |
| Hypoglycemia; n (%) | 15,124 (19.0%) | 15,177 (19.1%) | 0.00 | 9,112 (23.3%) | 9,024 (23.0%) | 0.01 | 19,509 (22.6%) | 19,400 (22.5%) | 0.00 |
| Hyperglycemia/DKA/HONK; n (%) | 42,188 (53.1%) | 41,857 (52.6%) | 0.001 | 21,187 (54.1%) | 21,055 (53.8%) | 0.01 | 42,600 (49.4%) | 42,539 (49.4%) | 0.00 |
| Skin infections; n (%) | 8,090 (10.2%) | 8,056 (10.1%) | 0.00 | 4,104 (10.5%) | 3,990 (10.2%) | 0.01 | 8,541 (9.9%) | 8,566 (9.9%) | 0.00 |
| Other comorbidities | | | | | | | | | |
| Microalbuminuria or proteinuria; n (%) | 3,911 (4.9%) | 3,842 (4.8%) | 0.00 | 2,342 (6.0%) | 2,317 (5.9%) | 0.00 | 4,556 (5.3%) | 4,452 (5.2%) | 0.01 |
| Acute kidney injury; n (%) | 5,238 (6.6%) | 5,236 (6.6%) | 0.00 | 2,165 (5.5%) | 2,087 (5.3%) | 0.01 | 3,729 (4.3%) | 3,762 (4.4%) | 0.00 |
| CKD stage 1–2; n (%) | 3,603 (4.5%) | 3,549 (4.5%) | 0.00 | 1,410 (3.6%) | 1,409 (3.6%) | 0.00 | 3,521 (4.1%) | 3,583 (4.2%) | 0.00 |
| CKD stage 3–4; n (%) | 11,173 (14.1%) | 11,117 (14.0%) | 0.00 | 4,366 (11.2%) | 4,287 (10.9%) | 0.01 | 9,317 (10.8%) | 9,238 (10.7%) | 0.00 |
| Unspecified CKD; n (%) | 4,472 (5.6%) | 4,462 (5.6%) | 0.00 | 1,953 (5.0%) | 1,912 (4.9%) | 0.01 | 3,491 (4.1%) | 3,479 (4.0%) | 0.00 |
| Urinary tract infections; n (%) | 10,771 (13.5%) | 10,818 (13.6%) | 0.00 | 4,630 (11.8%) | 4,625 (11.8%) | 0.00 | 10,137 (11.8%) | 10,058 (11.7%) | 0.00 |
| COPD; n (%) | 8,914 (11.2%) | 8,911 (11.2%) | 0.00 | 3,906 (10.0%) | 3,846 (9.8%) | 0.01 | 7,860 (9.1%) | 7,909 (9.2%) | 0.00 |
| Asthma; n (%) | 8,083 (10.2%) | 8,050 (10.1%) | 0.00 | 4,357 (11.1%) | 4,290 (11.0%) | 0.01 | 10,140 (11.8%) | 10,145 (11.8%) | 0.00 |
| Obstructive sleep apnea; n (%) | 20,290 (25.5%) | 20,293 (25.5%) | 0.00 | 11,636 (29.7%) | 11,597 (29.6%) | 0.00 | 28,394 (32.9%) | 28,358 (32.9%) | 0.00 |
| Serious bacterial infections; n (%) | 2,798 (3.5%) | 2,804 (3.5%) | 0.00 | 1,140 (2.9%) | 1,106 (2.8%) | 0.01 | 1,983 (2.3%) | 1,973 (2.3%) | 0.00 |
| Pneumonia; n (%) | 4,006 (5.0%) | 4,048 (5.1%) | 0.00 | 1,604 (4.1%) | 1,576 (4.0%) | 0.00 | 3,133 (3.6%) | 3,114 (3.6%) | 0.00 |
| Liver disease; n (%) | 11,550 (14.5%) | 11,473 (14.4%) | 0.00 | 6,351 (16.2%) | 6,236 (15.9%) | 0.01 | 14,548 (16.9%) | 14,477 (16.8%) | 0.00 |
| MASH/MASLD; n (%) | 6,375 (8.0%) | 6,278 (7.9%) | 0.01 | 3,692 (9.4%) | 3,630 (9.3%) | 0.01 | 9,214 (10.7%) | 9,152 (10.6%) | 0.00 |
| Osteoarthritis; n (%) | 21,376 (26.9%) | 21,391 (26.9%) | 0.00 | 10,325 (26.4%) | 10,334 (26.4%) | 0.00 | 23,274 (27.0%) | 23,372 (27.1%) | 0.00 |
| Depression; n (%) | 15,261 (19.2%) | 15,175 (19.1%) | 0.00 | 7,749 (19.8%) | 7,679 (19.6%) | 0.00 | 17,010 (19.7%) | 17,046 (19.8%) | 0.00 |
| Dementia; n (%) | 2,955 (3.7%) | 2,909 (3.7%) | 0.00 | 1,053 (2.7%) | 1,041 (2.7%) | 0.00 | 1,661 (1.9%) | 1,662 (1.9%) | 0.00 |
| Delirium or psychosis; n (%) | 1,218 (1.5%) | 1,205 (1.5%) | 0.00 | 531 (1.4%) | 510 (1.3%) | 0.01 | 883 (1.0%) | 875 (1.0%) | 0.00 |
| Anxiety; n (%) | 13,744 (17.3%) | 13,792 (17.3%) | 0.00 | 8,427 (21.5%) | 8,179 (20.9%) | 0.02 | 19,929 (23.1%) | 19,771 (22.9%) | 0.00 |
| Sleep disorders; n (%) | 24,589 (30.9%) | 24,449 (30.8%) | 0.00 | 11,988 (30.6%) | 11,931 (30.5%) | 0.00 | 28,221 (32.7%) | 28,233 (32.8%) | 0.00 |
| Anemia; n (%) | 14,705 (18.5%) | 14,618 (18.4%) | 0.00 | 6,526 (16.7%) | 6,466 (16.5%) | 0.00 | 14,226 (16.5%) | 14,207 (16.5%) | 0.00 |
| COVID; n (%) | 4,830 (6.1%) | 5,015 (6.3%) | 0.01 | 4,697 (12.0%) | 4,693 (12.0%) | 0.00 | 9,847 (11.4%) | 9,944 (11.5%) | 0.00 |
| Hyperthyroidism/other thyroid disorders; n (%) | 19,154 (24.1%) | 19,121 (24.1%) | 0.00 | 9,209 (23.5%) | 9,094 (23.2%) | 0.01 | 21,679 (25.2%) | 21,772 (25.3%) | 0.00 |
| Hypothyroidism; n (%) | 15,441 (19.4%) | 15,392 (19.4%) | 0.00 | 7,418 (18.9%) | 7,277 (18.6%) | 0.01 | 17,371 (20.2%) | 17,404 (20.2%) | 0.00 |
| Urinary incontinence; n (%) | 3,890 (4.9%) | 3,811 (4.8%) | 0.01 | 1,823 (4.7%) | 1,821 (4.7%) | 0.00 | 3,909 (4.5%) | 3,867 (4.5%) | 0.00 |
| Use of other medications | | | | | | | | | |
| Metformin; n (%) | 47,654 (59.9%) | 47,488 (59.7%) | 0.00 | 20,297 (51.8%) | 20,367 (52.0%) | 0.00 | 42,087 (48.8%) | 42,204 (49.0%) | 0.00 |
| Insulins; n (%) | 14,450 (18.2%) | 14,038 (17.7%) | 0.01 | 8,197 (20.9%) | 8,115 (20.7%) | 0.01 | 13,932 (16.2%) | 13,906 (16.1%) | 0.00 |
| Sulfonylureas; n (%) | 18,417 (23.2%) | 18,366 (23.1%) | 0.00 | 6,974 (17.8%) | 6,911 (17.7%) | 0.00 | 11,495 (13.3%) | 11,418 (13.2%) | 0.00 |
| SGLT2 inhibitors; n (%) | 12,234 (15.4%) | 11,864 (14.9%) | 0.01 | 8,143 (20.8%) | 8,161 (20.8%) | 0.00 | 15,897 (18.4%) | 15,928 (18.5%) | 0.00 |
| Any other glucose-lowering drugs; n (%) | 4,865 (6.1%) | 4,835 (6.1%) | 0.00 | 2,316 (5.9%) | 2,307 (5.9%) | 0.00 | 4,370 (5.1%) | 4,384 (5.1%) | 0.00 |
| ACE or ARB; n (%) | 59,870 (75.3%) | 59,907 (75.4%) | 0.00 | 28,583 (73.0%) | 28,472 (72.7%) | 0.01 | 61,304 (71.1%) | 61,148 (70.9%) | 0.00 |
| ARNI; n (%) | 864 (1.1%) | 861 (1.1%) | 0.00 | 627 (1.6%) | 624 (1.6%) | 0.00 | 1,347 (1.6%) | 1,378 (1.6%) | 0.00 |
| Beta-blockers; n (%) | 33,255 (41.8%) | 33,333 (41.9%) | 0.00 | 15,200 (38.8%) | 15,182 (38.8%) | 0.00 | 32,384 (37.6%) | 32,591 (37.8%) | 0.01 |
| Calcium channel blockers; n (%) | 26,321 (33.1%) | 26,403 (33.2%) | 0.00 | 12,321 (31.5%) | 12,286 (31.4%) | 0.00 | 26,196 (30.4%) | 26,243 (30.4%) | 0.00 |
| Loop diuretics; n (%) | 13,878 (17.5%) | 13,927 (17.5%) | 0.00 | 6,147 (15.7%) | 6,040 (15.4%) | 0.01 | 12,882 (14.9%) | 12,893 (15.0%) | 0.00 |

**Table 1 (continued) | Baseline characteristics of patients initiating semaglutide versus sitagliptin, tirzepatide versus dulaglutide and tirzepatide versus semaglutide after 1:1 propensity score matching**

| | n (%) | | | | | | | | |
|---|---|---|---|---|---|---|---|---|---|
| | **Semaglutide versus sitagliptin** | | | **Tirzepatide versus dulaglutide** | | | **Tirzepatide versus semaglutide** | | |
| | Semaglutide (n = 79,501) | Sitagliptin (n = 79,501) | SMD | Tirzepatide (n = 39,152) | Dulaglutide (n = 39,152) | SMD | Tirzepatide (n = 86,191) | Semaglutide (n = 86,191) | SMD |
| Thiazides; n (%) | 27,881 (35.1%) | 27,961 (35.2%) | 0.00 | 13,220 (33.8%) | 13,261 (33.9%) | 0.00 | 30,433 (35.3%) | 30,394 (35.3%) | 0.00 |
| Other diuretics; n (%) | 6,442 (8.1%) | 6,421 (8.1%) | 0.00 | 3,347 (8.5%) | 3,342 (8.5%) | 0.00 | 7,464 (8.7%) | 7,526 (8.7%) | 0.00 |
| Nitrates; n (%) | 5,322 (6.7%) | 5,234 (6.6%) | 0.00 | 2,199 (5.6%) | 2,202 (5.6%) | 0.00 | 4,173 (4.8%) | 4,135 (4.8%) | 0.00 |
| Statins; n (%) | 63,368 (79.7%) | 63,301 (79.6%) | 0.00 | 31,016 (79.2%) | 31,109 (79.5%) | 0.01 | 64,885 (75.3%) | 64,817 (75.2%) | 0.00 |
| PCSK9 inhibitors/other lipid-lowering drugs; n (%) | 10,492 (13.2%) | 10,370 (13.0%) | 0.01 | 5,262 (13.4%) | 5,154 (13.2%) | 0.01 | 11,539 (13.4%) | 11,571 (13.4%) | 0.00 |
| Antiplatelet medications; n (%) | 8,847 (11.1%) | 8,786 (11.1%) | 0.00 | 3,764 (9.6%) | 3,723 (9.5%) | 0.00 | 7,159 (8.3%) | 7,239 (8.4%) | 0.00 |
| Oral anticoagulants; n (%) | 8,007 (10.1%) | 8,055 (10.1%) | 0.00 | 3,538 (9.0%) | 3,540 (9.0%) | 0.00 | 7,402 (8.6%) | 7,426 (8.6%) | 0.00 |
| NSAIDs; n (%) | 26,168 (32.9%) | 26,171 (32.9%) | 0.00 | 12,818 (32.7%) | 12,802 (32.7%) | 0.00 | 29,544 (34.3%) | 29,645 (34.4%) | 0.00 |
| COPD/asthma medications; n (%) | 26,536 (33.4%) | 26,463 (33.3%) | 0.00 | 13,489 (34.5%) | 13,344 (34.1%) | 0.01 | 30,956 (35.9%) | 30,907 (35.9%) | 0.00 |
| Urinary tract infections antibiotics; n (%) | 36,563 (46.0%) | 36,475 (45.9%) | 0.00 | 17,746 (45.3%) | 17,734 (45.3%) | 0.00 | 40,944 (47.5%) | 40,896 (47.4%) | 0.00 |
| Healthcare utilization | | | | | | | | | |
| Number of endocrinologist visits; mean (s.d.) | 0.43 (1.20) | 0.41 (1.54) | 0.02 | 0.44 (1.31) | 0.44 (1.43) | 0.01 | 0.42 (1.34) | 0.42 (1.37) | 0.00 |
| Number of cardiologist visits; mean (s.d.) | 1.46 (3.37) | 1.45 (3.31) | 0.01 | 1.29 (2.97) | 1.30 (3.16) | 0.00 | 1.28 (2.99) | 1.29 (3.07) | 0.00 |
| Number of hospitalizations; mean (s.d.) | 2.54 (21.95) | 2.42 (17.31) | 0.01 | 1.81 (13.73) | 1.75 (14.13) | 0.00 | 1.22 (10.95) | 1.26 (16.98) | 0.00 |
| Emergency department visit; n (%) | 26,726 (33.6%) | 26,693 (33.6%) | 0.00 | 11,738 (30.0%) | 11,570 (29.6%) | 0.01 | 22,894 (26.6%) | 23,039 (26.7%) | 0.00 |
| Burden of comorbidities | | | | | | | | | |
| Combined comorbidity index; mean (s.d.) | 1.85 (2.39) | 1.84 (2.37) | 0.01 | 1.70 (2.26) | 1.68 (2.22) | 0.01 | 1.53 (2.13) | 1.54 (2.13) | 0.01 |
| Claims frailty index; mean (s.d.) | 0.17 (0.06) | 0.17 (0.06) | 0.00 | 0.16 (0.06) | 0.16 (0.06) | 0.00 | 0.15 (0.05) | 0.15 (0.05) | 0.00 |
| Healthy behavior markers | | | | | | | | | |
| Colonoscopy/sigmoidoscopy; n (%) | 7,905 (9.9%) | 7,919 (10.0%) | 0.00 | 4,034 (10.3%) | 4,047 (10.3%) | 0.00 | 9,377 (10.9%) | 9,251 (10.7%) | 0.01 |
| Flu/pneumococcal vaccine; n (%) | 23,902 (30.1%) | 23,969 (30.1%) | 0.00 | 12,399 (31.7%) | 12,410 (31.7%) | 0.00 | 24,895 (28.9%) | 24,898 (28.9%) | 0.00 |
| Pap smear; n (%) | 5,817 (7.3%) | 5,878 (7.4%) | 0.00 | 3,016 (7.7%) | 3,061 (7.8%) | 0.00 | 8,081 (9.4%) | 8,064 (9.4%) | 0.00 |
| Mammograms; n (%) | 19,056 (24.0%) | 19,162 (24.1%) | 0.00 | 9,746 (24.9%) | 9,830 (25.1%) | 0.01 | 23,658 (27.4%) | 23,538 (27.3%) | 0.00 |
| Telemedicine; n (%) | 17,215 (21.7%) | 17,097 (21.5%) | 0.00 | 9,743 (24.9%) | 9,719 (24.8%) | 0.00 | 21,862 (25.4%) | 21,958 (25.5%) | 0.00 |
| Laboratory and diagnostic tests | | | | | | | | | |
| HbA1c tests; mean (s.d.) | 2.39 (1.50) | 2.38 (1.84) | 0.00 | 2.41 (1.38) | 2.41 (1.32) | 0.00 | 2.27 (1.35) | 2.26 (1.35) | 0.01 |
| Lipid panels; mean (s.d.) | 1.66 (1.32) | 1.66 (1.54) | 0.01 | 1.57 (1.15) | 1.57 (1.18) | 0.00 | 1.63 (1.15) | 1.63 (1.20) | 0.00 |

ACE, angiotensin-converting enzyme inhibitor; ARB, angiotensin receptor blocker; ARNI, angiotensin receptor/neprilysin inhibitor; COPD, chronic obstructive pulmonary disease; DKA, diabetic ketoacidosis; HbA1c, hemoglobin A1c; HONK, hyperglycemic hyperosmolar nonketotic state; MASH, metabolic dysfunction-associated steatohepatitis; MASLD, metabolic dysfunction-associated steatotic liver disease; n, number of individuals; NSAIDs, nonsteroidal anti-inflammatory drugs; PCSK9, proprotein convertase subtilisin/kexin type 9; SMD, standardized mean difference; TIA, transient ischemic attack.

## Subgroups and sensitivity analyses

Prespecified subgroup analyses in the expanded populations for age showed no meaningful treatment effect heterogeneity (Fig. 4 and Supplementary Table 24). Across sex subgroups, effects were similar for semaglutide versus sitagliptin and for tirzepatide versus semaglutide whereas for tirzepatide versus dulaglutide, estimates suggested male patients to benefit more from tirzepatide than female patients. Among patients receiving concomitant sodium–glucose cotransporter-2 (SGLT2) inhibitors at baseline, no meaningful difference to those patients without concomitant SGLT2 inhibitor use was observed.

Sensitivity analyses using an as-started causal contrast led to modestly attenuated estimates. Effect estimates remained robust when restricting to patients with available hemoglobin A1c and including it in the propensity score (Supplementary Table 23).

On post hoc analyses, we further evaluated the comparative effectiveness of semaglutide versus dulaglutide in the expanded population to contextualize whether a reduction in the primary endpoint, relative to the comparisons including tirzepatide, was similar

or greater. Effect estimates confirmed an increased risk (HR 1.24, 95% CI 1.15 to 1.34), which further supported the robustness of our findings. The comparative effectiveness of an expanded 2-year on-treatment analysis that followed patients who stay on the exposure or comparator therapies for a prolonged time showed concordant results with the 1-year on-treatment analysis (Supplementary Table 25).

## Discussion

In this database study, treatment with semaglutide (against sitagliptin, a placebo proxy) led to a reduced risk of MACE, and tirzepatide (against dulaglutide) demonstrated a similar risk reduction in patients at elevated cardiovascular risk with obesity and type 2 diabetes. These findings align closely with prior cardiovascular outcome trials and solidify the evidence base by demonstrating consistent benefits in broader patient populations treated in clinical practice.

In the direct comparison of tirzepatide versus semaglutide, we observed CIs compatible with no difference and point estimates indicating a modest numeric advantage of semaglutide, if any, for reducing

**Table 2 | Benchmarking of results from trial emulations against reference trials to inform analyses in expanded populations**

| | Endpoint | Trial emulation estimate (95% CI) | Trial estimate (95% CI) | SA | DA | EA | SD |
|---|---|---|---|---|---|---|---|
| SUSTAIN-6 | MACE | 0.68 (0.60 to 0.77) | 0.74 (0.58 to 0.95) | Yes | Yes | Yes | Yes |
| | Myocardial infarction | 0.70 (0.57 to 0.86) | 0.74 (0.51 to 1.08) | N/A[a] | Yes | Yes | Yes |
| | Stroke | 0.82 (0.65 to 1.03) | 0.61 (0.38 to 0.99) | N/A[a] | Yes | Yes | Yes |
| | All-cause mortality | 0.58 (0.48 to 0.71) | 1.05 (0.74 to 1.50) | N/A[a] | No | No | No |
| SURPASS-CVOT | MACE | 0.83 (0.69 to 1.01) | 0.92 (0.83 to 1.01) | Yes | Yes | Yes | Yes |
| | Myocardial infarction | 0.81 (0.61 to 1.06) | 0.86 (0.74 to 1.00) | N/A[a] | Yes | Yes | Yes |
| | Stroke | 0.92 (0.65 to 1.29) | 0.91 (0.76 to 1.09) | N/A[a] | Yes | Yes | Yes |
| | All-cause mortality | 0.76 (0.52 to 1.11) | 0.84 (0.75 to 0.94) | N/A[a] | Yes | Yes | Yes |

[a]Statistical agreement was assessed only for the primary endpoints. See details in the main text. DA, directional agreement; EA, estimate agreement; N/A, not applicable; SA, statistical agreement; SD, standardized difference agreement.

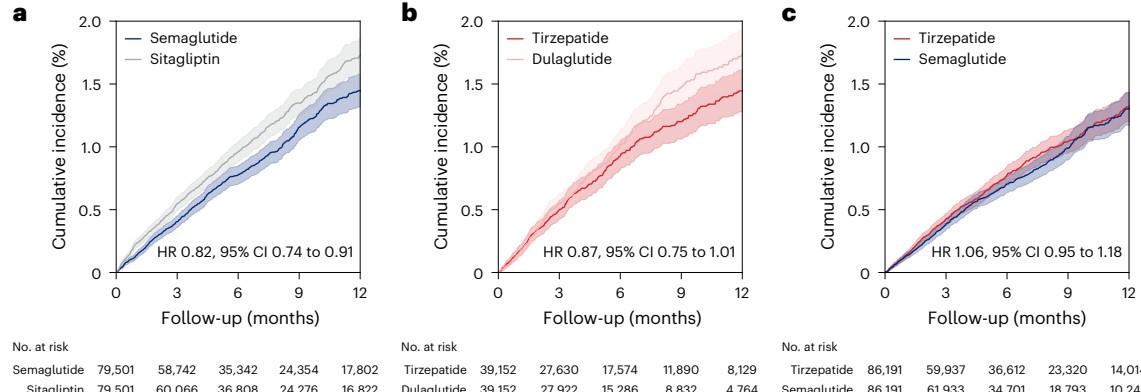

**Fig. 2 | Cumulative incidence curves for the composite endpoint of MACEs in expanded populations. a–c**, The composite endpoint of myocardial infarction or stroke in patients initiating semaglutide versus sitagliptin (**a**); the composite endpoint of myocardial infarction, stroke or all-cause mortality in patients initiating tirzepatide versus dulaglutide (**b**); and the composite endpoint of myocardial infarction, stroke or all-cause mortality in patients initiating tirzepatide versus semaglutide (**c**). The shaded bands represent 95% CIs around the Kaplan–Meier estimate.

MACE, particularly among patients at high cardiovascular risk. Conversely, point estimates for tirzepatide showed a potential modest advantage for heart failure endpoints, with CIs compatible with no difference, consistent with recent data supporting protective effects of semaglutide and tirzepatide on heart failure outcomes and a potential incremental benefit with the latter[22]. Potential explanations include the dual receptor agonism of tirzepatide[23]. Current insight into the cardiovascular biology of GIP is limited, and findings in preclinical and clinical studies point in different directions, ranging from potentially cardioprotective effects to heightened vascular inflammation and artherogenesis[24,25]. This aligns with findings from SURPASS-CVOT, which showed no meaningful additive effect on MACE compared to an older GLP-1 receptor agonist, and with findings of a potential greater protection in heart failure that could be compatible with more pronounced weight loss mediated by GIP[16,22]. In contrast, GLP-1 receptor activation has consistently reduced MACE in randomized trials[26,27]. Tirzepatide binds the GLP-1 receptor with lower affinity than semaglutide and exhibits distinct signaling bias, which may result in comparatively different downstream GLP-1 receptor signaling dynamics[23]. Furthermore, tirzepatide has shown longer titration periods in clinical practice, delaying attainment of full maintenance doses and potentially dampening early cardiovascular benefits[28]. These hypotheses require further confirmation.

Our findings offer comprehensive real-world evidence for the cardiovascular effectiveness of tirzepatide in clinical practice in the absence of studies with direct comparisons to semaglutide, and ahead of further evidence from the SURPASS-CVOT randomized controlled trial that compared tirzepatide against an older GLP-1 receptor agonist that is not frequently used in routine care[14,17]. As cardiovascular risk remains high among adults with type 2 diabetes and excess weight, the timely evaluation of new therapies is a public health priority. While randomized trials are the reference methodology for establishing treatment effectiveness, they leave many clinically relevant questions unanswered, which may delay access for new indications[29–31]. Nonrandomized database studies have inherent limitations; however, when rigorously designed using proven analytic approaches to emulate reference trials, real-world evidence of several glucose-lowering drug classes have demonstrated strong concordance with trial estimates[20,22,32]. As dozens of novel agents currently under study seek new indications, the question arises of whether the traditional practice of two confirmatory trials for every indication expansion remains justified, especially when database study emulations that benchmark against previously conducted, closely related reference trials have yielded aligned results[33,34].

This study demonstrates how database studies rooted in benchmarks against previously conducted randomized trials for the drugs of interest can produce complementary evidence to support expanded cardiovascular indications. By preregistering detailed protocols with contemporaneously documented amendments, we ensured methodological transparency. By aligning key protocol components and analytic frameworks with those of SUSTAIN-6 and SURPASS-CVOT, we produced self-critical evidence that allowed us to directly benchmark results against reference trials to inform subsequent analyses in

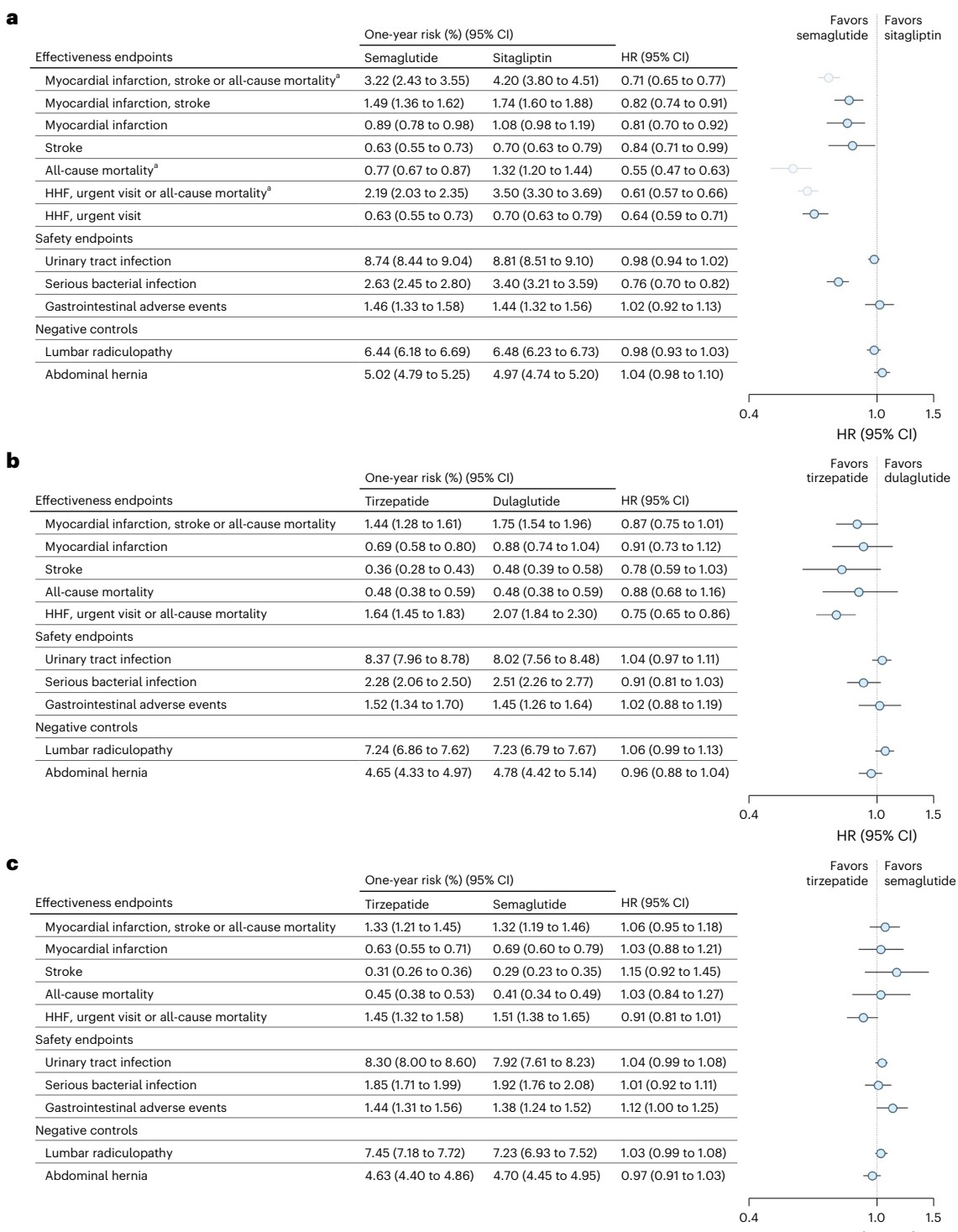

**Fig. 3 | Effectiveness and safety endpoints in expanded populations.**
**a**–**c**, The 1-year risks and HRs for effectiveness and safety endpoints, together with negative controls, are shown for 159,002 patients initiating semaglutide versus sitagliptin (**a**), 78,304 patients initiating tirzepatide versus dulaglutide (**b**) and 172,382 patients initiating tirzepatide versus semaglutide (**c**). [a]For the expanded population of patients initiating semaglutide versus sitagliptin in **a**, our analysis focused on endpoints that did not include all-cause mortality. See the main text for details. HHF, hospitalization for heart failure.

expanded populations. Close agreement observed between trials and estimates from the database analyses for all endpoints except one supported the fitness of the design and data for assessing cardiovascular effectiveness and safety. However, benchmarking against SUSTAIN-6 flagged disagreement for the effect on all-cause mortality, highlighting potential residual confounding for this secondary endpoint that could reflect preferential prescribing in patients for whom clinicians

anticipated limited life expectancy. The observed divergence in mortality results when comparing injectable semaglutide to sitagliptin (a placebo proxy) in benchmarking analyses led us to view the results for outcomes containing mortality with skepticism and focused our interpretation on the nonmortality clinical endpoints. For transparency we still reported the mortality findings. In contrast, the concordance of benchmarking results for all outcomes, including mortality, in the

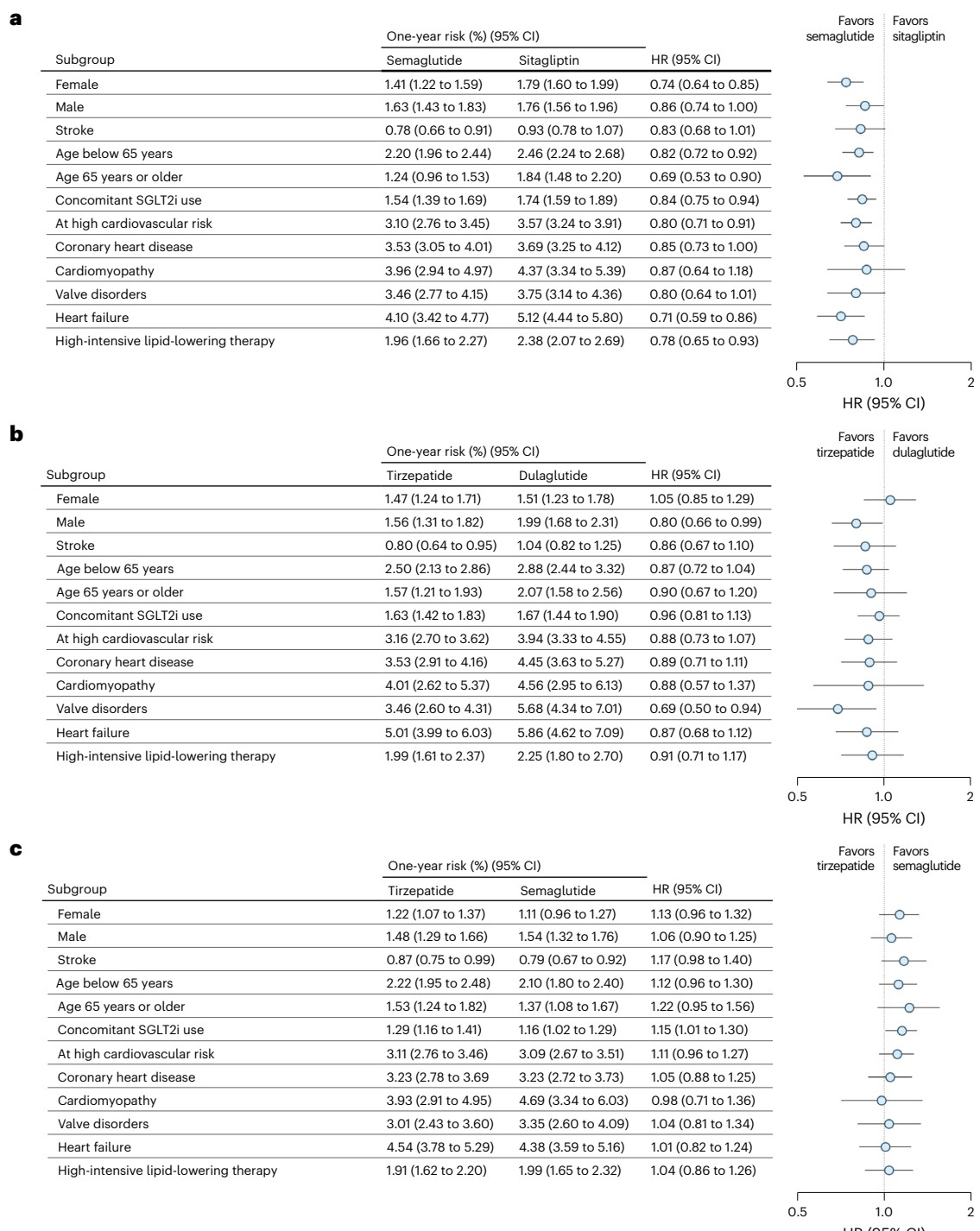

**Fig. 4 | MACEs in subgroups. a–c**, The 1-year risks and HRs for the composite endpoint of myocardial infarction or stroke in 159,002 patients initiating semaglutide versus sitagliptin (**a**); the composite endpoint of myocardial infarction, stroke or all-cause mortality in 78,304 patients initiating tirzepatide versus dulaglutide (**b**); and the composite endpoint of myocardial infarction, stroke or all-cause mortality in 172,382 patients initiating tirzepatide versus semaglutide (**c**). SGLT2i, SGLT2 inhibitor.

SURPASS-CVOT emulation study comparing tirzepatide to dulaglutide provided support for the validity of evaluating these outcomes in expanded study populations. As the landscape of cardiometabolic therapeutics evolves rapidly, real-world evidence may serve as a critical tool to generate comparative insights beyond trials that is essential for clinical decision-making and regulatory evaluation.

This study has several limitations. First, treatment allocation was not randomized, raising the potential for residual confounding despite extensive pretreatment covariate adjustment through propensity score matching. The resulting concordance between the emulated and actual SUSTAIN-6 and SURPASS-CVOT trial estimates strengthen confidence in the internal validity of our findings for the successfully assessed endpoints. When differences in results for endpoints were observed, this informed subsequent analyses. Second, information on outcomes, comorbidities and cardiovascular risk factors was derived from administrative claims, which may be less reliable than

trial-based assessments. We addressed this by incorporating a range of algorithms based on diagnosis, procedure and prescription claims, as well as frailty indicators and health service utilization. The endpoint algorithms were validated and showed a sensitivity of over 99% for mortality in the National Death Index, a positive predictive value of 94% for myocardial infarction, 95% for stroke and 98% for heart failure hospitalization[33,35–40]. Third, medication exposure was identified from pharmacy dispensing records, which is more accurate than prescribing information. Given the modest treatment persistence observed in clinical practice, our results may not capture long-term cardiovascular outcomes. The divergence in event rates between treatment groups observed within 1 year across both randomized trials and equally in our analyses suggests that meaningful effects may emerge within a short time frame. Fourth, although our data represent a diverse population, findings may best apply to the USA and have more limited applicability to international settings. Last, we assumed neutral effects on cardiovascular endpoints for sitagliptin as a comparator in the semaglutide analysis. This assumption is supported by a prior outcome trial and observational data[20,41].

Relevant for clinical practice, our findings show that treatment with semaglutide lowered the risk of MACE compared to sitagliptin, while tirzepatide showed at least comparable benefit to dulaglutide, an older GLP-1 receptor agonist with established cardiovascular efficacy. In a direct head-to-head comparison, tirzepatide demonstrated similar benefits in reducing MACE as semaglutide. These findings provide timely insights into the cardiovascular effectiveness of modern GLP-1 receptor agonist-based therapies that can inform clinical decision-making in the absence of a head-to-head randomized trial.

## Online content

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

## Methods

### Data sources

The five cohort studies were conducted using three nationwide claims databases, including Medicare Parts A, B and D (2018 through 2020), Optum Clinformatics Data Mart (2018 through February 2025), and Merative MarketScan (2018 through 2023). Medicare claims data includes beneficiaries aged 65 years and older enrolled in traditional fee for service. Optum and MarketScan databases capture commercially insured individuals across the USA. All three databases capture de-identified, longitudinal patient-level information on demographics; diagnoses and procedures from inpatient, outpatient and emergency department encounters; and prescription medications dispensed to outpatients. Each database permits tracking of healthcare utilization and medication exposure over time, and patients may contribute to more than one database if they meet the respective eligibility criteria.

### Specification and emulation of the studies

To inform our study design and analytic approach for the head-to-head tirzepatide versus semaglutide comparison of interest, we sought to conduct studies that allowed us to benchmark results of similarly designed database studies against results from randomized trials asking closely related questions. Insights from benchmarking can inform and lead to changes in subsequent analyses for the expanded questions of interest. We emulated and benchmarked against two cardiovascular outcome trials (reference trials), SUSTAIN-6 (semaglutide versus placebo) and SURPASS-CVOT (tirzepatide versus dulaglutide) in a pair of cohort studies. Key protocol components, including eligibility criteria, treatment strategies, assignment procedures, follow-up, outcome definitions, causal contrast, identifying assumptions and the data analysis plan, were translated into operational definitions using validated claims-based algorithms and observational analogs of key study design choices.

SUSTAIN-6 evaluated once-weekly semaglutide versus placebo in patients at moderate and high cardiovascular risk with type 2 diabetes, showing a 26% relative risk reduction in MACE (HR 0.74, 95% CI 0.58 to 0.95). SURPASS-CVOT compared once-weekly tirzepatide with dulaglutide in adults with type 2 diabetes and high cardiovascular risk, showing a 9% relative risk reduction in MACE (HR 0.92, 95% CI 0.83 to 1.01).

Building on these reference trials, we specified two protocols that expanded eligibility to assess the effectiveness of semaglutide versus sitagliptin and tirzepatide versus dulaglutide in populations reflective of patients routinely seen in clinical practice, spanning low, moderate or high cardiovascular risk (Supplementary Table 26).

Finally, we specified a protocol for a direct head-to-head comparison of tirzepatide versus semaglutide. These three protocols were emulated using the same analytic approach applied in the trial emulations used for benchmarking. Specification and emulation of the reference trial including our expansion studies are stated in the Supplementary Information, following the TrAnsparent ReportinG of observational studies Emulating a Target trial (TARGET) guideline (Supplementary Tables 27–29)[42].

### Transparency statement

We prespecified and registered the study protocols for the five cohort studies before outcome analyses were conducted to enhance transparency and minimize analytical bias. Protocols and their amendments are publicly accessible in ClinicalTrials.gov under the National Clinical Trial (NCT) identifiers NCT06659744, NCT07088718 and NCT07096063, and in the Open Science Framework (osf.io/38rw9)[43]. Each protocol outlines the study rationale and design following the HARmonized Protocol Template to Enhance Reproducibility (HARPER), a structured framework for transparent and reproducible observational study design[44]. The study was conducted between October 2024 and August 2025 and reported following the Reporting of Studies Conducted Using Observational Routinely Collected Health Data for Pharmacoepidemiology (RECORD-PE) statement and the Strengthening the Reporting of Observational Studies in Epidemiology (STROBE) reporting guidance[45].

### Study cohorts

We designed five active-comparator, new-user cohort studies including patients with obesity and type 2 diabetes at elevated cardiovascular risk[46]. All patients were required to have continuous enrollment of at least 12 months before cohort entry and meet eligibility criteria for one of the five cohort studies defined below (study design diagrams depicted in Supplementary Figs. 1–3).

Semaglutide versus sitagliptin: in one set of analyses, we compared initiators of semaglutide with initiators of sitagliptin in patients at elevated cardiovascular risk with type 2 diabetes and obesity. Sitagliptin, a dipeptidyl peptidase-4 inhibitor, was selected as an active-comparator placebo proxy because it was recommended in clinical guidelines as a second-line glucose-lowering therapy, had a similar high cost to the study drugs, showed no effect on cardiovascular outcomes in a large randomized controlled trial and has been validated as a placebo proxy in prior claims-based research[20,41]. This choice was intended to avoid potential confounding associated with non-user comparisons and other active comparators that have been linked to a possible excess risk or decrease in risk of major adverse cardiovascular outcomes[27,47].

To benchmark our findings against randomized evidence, we first applied eligibility criteria after SUSTAIN-6 that required patients to be at moderate or high cardiovascular risk[11]. Moderate risk was defined as age ≥60 years with subclinical cardiovascular disease, such as microalbuminuria or proteinuria, left ventricular hypertrophy or dysfunction, or an ankle–brachial index <0.9. High risk was defined as age ≥50 years with established cardiovascular disease, including prior major atherosclerotic events, revascularization, substantial arterial stenosis, ischemia, New York Heart Association class II–III heart failure or CKD.

We then expanded the cohort by relaxing the trial eligibility criteria to reflect populations typically encountered in practice at low, moderate or high cardiovascular risk, including patients on lipid-lowering therapy or antihypertensive treatment (Fig. 1 and Supplementary Information).

Tirzepatide versus dulaglutide: using the same framework, we compared initiators of tirzepatide with initiators of dulaglutide to emulate the design and rationale of the SURPASS-CVOT trial[17]. Patients enrolled were at high cardiovascular risk. At the time the protocol was finalized, trial results were not yet available; the study therefore aimed at predicting eventual trial readouts, countering the potential criticism of tailoring the design toward known results. While conducting the analysis, the primary results became available and allowed us to benchmark against known outcomes[16]. After benchmarking, we expanded the cohort to evaluate effects in broader patient populations at low, moderate or high cardiovascular risk (Fig. 1).

Tirzepatide versus semaglutide: upon establishing agreement with the trial benchmarks, we conducted a direct comparison of new initiators of tirzepatide and semaglutide to assess differences in cardiovascular and select safety outcomes in clinical practice (Fig. 1).

### Outcomes

The primary endpoint was MACE, a composite of all-cause mortality, myocardial infarction or stroke. Secondary outcomes included the individual components of the primary endpoint. For the expanded populations, we further assessed a composite of hospitalization for heart failure, all-cause mortality or an urgent visit for heart failure requiring intravenous diuretics, as well as select safety endpoints including gastrointestinal adverse events, serious bacterial infections and urinary tract infections[48]. Endpoints were ascertained during 52 weeks of follow-up in an as-treated approach, with censoring at the first occurrence of an endpoint, treatment discontinuation (plus 45-day

grace period), treatment switching, start of another agent within the same class, disenrollment or study end.

Insights from the benchmarking studies prompted an amendment to the protocol for the comparison of semaglutide versus sitagliptin in expanded populations. Owing to the observed divergence in the results for the mortality endpoint between the SUSTAIN-6 benchmarking study and the SUSTAIN-6 trial findings, the result of the all-cause mortality endpoint was viewed skeptically. Additional composite endpoints with the mortality component removed were added to the protocol comparing semaglutide to sitagliptin in the expanded population in amendments documented on ClinicalTrials.gov. Given the observed concordance in results for the SURPASS-CVOT benchmarking study and SURPASS-CVOT trial findings, no amendments were made to the protocol for the study that expanded the population for comparisons of tirzepatide versus dulaglutide. With the expectation that the confounding structure for the head-to-head comparison would more closely resemble that of the SURPASS-CVOT benchmarking study, no amendments were made to the protocol for the head-to-head comparison of tirzepatide and semaglutide.

### Subgroups, sensitivity and post hoc analyses

Subgroups of interest in the expanded populations included stratifying by age (<65 or ≥65 years), sex, concomitant use of SGLT2 inhibitors (yes or no) and patients at high cardiovascular risk. In addition, we assessed the effectiveness in patients with certain cardiovascular conditions, such as coronary heart disease, cardiomyopathies, valve disorders or heart failure, as well as patients under high-intensive lipid-lowering therapy.

Sensitivity analyses included as-started follow-up emulating a per-protocol analysis, restricting analyses to patients with hemoglobin A1c measurement in the past 120 days before initiating the study drugs and adjusting for the most recent readout. We included two negative control outcome analyses to evaluate potential residual confounding, defined as new diagnosis of lumbar radiculopathy and abdominal hernia, which have no biologically plausible association with the study drugs.

To contextualize whether the reduction in the primary composite endpoint, relative to comparisons involving tirzepatide, was similar or greater, we conducted a post hoc analysis evaluating the comparative effectiveness of semaglutide versus dulaglutide in the expanded population. Another post hoc analysis examined extending the on-treatment follow-up to 2 years among patients who remained on therapy to assess potential changes in effectiveness beyond the first year.

### Covariates

We evaluated a broad set of covariates to capture potential confounding. These included demographics (for example, age, sex, race and claims-based proxies for socioeconomic status such as geographic location and copayments), as well as detailed cardiovascular risk profiles. The latter encompassed both traditional risk factors (for example, smoking status, obesity severity, hypertension and hyperlipidemia) and established cardiovascular disease (for example, prior myocardial infarction, unstable or stable angina, ischemic stroke or transient ischemic attack, peripheral artery disease, prior coronary or peripheral revascularization, atrial fibrillation, cardiomyopathy, valvular heart disease, chronic heart failure and device implantation such as pacemakers or implantable cardioverter–defibrillators). We also captured subclinical indicators of elevated risk, such as microalbuminuria, edema or obstructive sleep apnea. Additional covariates included markers of cardiometabolic burden and diabetes-related complications (for example, nephropathy, neuropathy, retinopathy, diabetic foot ulcers and episodes of hypo- or hyperglycemia), renal disease (acute or CKD and hypertensive nephropathy), coexisting comorbidities (for example, chronic obstructive pulmonary disease, asthma, obstructive sleep apnea, depression and dementia) and infection

history (for example, pneumonia, urinary tract infection, COVID-19 and influenza). Medication history covered glucose-lowering therapies, cardiovascular drugs and other commonly prescribed agents. Healthcare utilization was described by hospitalizations (including heart failure-related admissions), emergency visits, specialist encounters, diagnostic testing and preventive care services. Where available, laboratory values and vitals were reported (for example, B-type natriuretic peptide, hemoglobin A1c, serum creatinine, estimated glomerular filtration rate, lipid profile, systolic blood pressure and BMI). Full definitions and assessment windows are provided in the Supplementary Information.

### Benchmarking against randomized trials and predefined binary agreement metrics

To assess concordance between our database emulations and their randomized trial counterparts, we compared the primary endpoint results with the published findings from SUSTAIN-6 and, once results became available, from SURPASS-CVOT[18,19,21]. Agreement between the trial and real-world evidence estimates was evaluated using four prespecified binary metrics defined by the RCT-DUPLICATE initiative:

(1) Statistical agreement, defined as both the database study and trial estimates and their 95% CIs lying on the same side of the null (assessed for primary endpoints only, secondary endpoints were not powered for statistical agreement)
(2) Directional agreement, defined as both the database study and trial estimates lying on the same side of the null
(3) Estimate agreement, defined as the estimate of the database study falling within the 95% CI of the trial
(4) Standardized difference agreement, defined as an absolute standardized difference $|Z| < 1.96$, with $Z = \frac{\hat{\theta}_{RCT} - \hat{\theta}_{RWE}}{\sqrt{\hat{\sigma}^2_{RCT} + \hat{\sigma}^2_{RWE}}}$, where $\hat{\theta}_{RCT}$ and $\hat{\theta}_{RWE}$ are the treatment effect estimates with associated variances $\hat{\sigma}^2$ (RWE, real-world evidence; RCT, randomized clinical trial)

### Statistical analysis

We summarized baseline covariates using appropriate descriptive statistics and assessed balance between groups with standardized mean differences, considering values below 0.10 indicative of adequate balance. Propensity scores for each pairwise comparison were estimated with logistic regression, using the variables described in 'Covariates' section. To mimic randomization in the database studies, we matched eligible patients who initiated each study drug in a 1:1 ratio to initiators of the comparator drug based on the propensity score, using a caliper width of 0.01 on the propensity score scale. Absolute risks at 52 weeks were derived from Kaplan–Meier estimates on the combined patient-level data from the databases as well as individually (results for the primary endpoints before and after propensity score matching are found in Supplementary Table 30). For individual components of the primary endpoint other than death, absolute risks were calculated using the Aalen–Johansen estimator to account for competing risks. Risk differences at 52 weeks were obtained as contrasts of these estimates. Pointwise 95% CIs were derived via a nonparametric patient-level bootstrap with 1,000 replicates, resampling patients with replacement from the analytic cohort. HRs and 95% CIs were calculated with Cox proportional hazards models. Database-specific estimates were pooled using a fixed-effects inverse variance meta-analysis.

Under the assumptions made in the power calculations (Supplementary Table 31), analyses of the primary endpoints in the emulation of SUSTAIN-6 and the comparison of semaglutide versus sitagliptin in expanded populations was estimated to have >99% power for superiority. For the benchmarking emulation of SURPASS-CVOT and the comparison of tirzepatide versus dulaglutide in expanded populations, the estimated power to detect noninferiority was >80%.

For the comparison of tirzepatide versus semaglutide in expanded populations, the estimated power to detect noninferiority was >90%. Analyses were performed with Python, R and the Aetion Evidence Platform, a validated system extensively used for reproducible real-world evidence studies and trial emulations, benchmarked against US Food and Drug Administration Sentinel Initiative workflows.

## Reporting summary

Further information on research design is available in the Nature Portfolio Reporting Summary linked to this article.

## Data availability

The study was approved by the Mass General Brigham Institutional Review Board. The use of de-identified secondary data qualified for a waiver of informed consent by US federal regulations. Data use agreements and licensing agreements do not allow sharing of patient-level claims data with third parties. However, data can be requested at the vendors directly (Optum Clinformatics, connected@optum.com; Medicare, resdac@umn.edu; Merative MarketScan, marketscan.support@merative.com). The analytical code with which to create the tables, figures and analysis results for this study is available via GitHub at https://github.com/nilskruger/Major-Adverse-Cardiovascular-Outcomes-for-Semaglutide-and-Tirzepatide-in-Clinical-Practice/.

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

## Acknowledgements

We like to thank N. Dimpfl for helpful comments. This work was funded by the National Institutes of Health (grant nos. R01-HL141505 and R01-AR080194) and the German Heart Foundation (grant nos. S/02/24 and SRF-HF/24).

## Author contributions

N.K. had full access to all of the data in the study and takes responsibility for the integrity of the data and the accuracy of the data analysis. Concept and design by N.K., S.S. and S.V.W. Acquisition, analysis or interpretation of data by all authors. Drafting of the manuscript by N.K. Critical review of the paper for important intellectual content by all authors. Statistical analysis by N.K., G.H. and S.V.W. Funding obtained by N.K., S.S. and S.V.W. Administrative, technical or material support from N.K., S.S. and S.V.W. Supervision by S.S. and S.V.W.

## Competing interests

S.S. reports personal fees from Aetion Inc., a software-enabled analytics company, and grants from Bayer, UCB and Boehringer Ingelheim to Brigham and Women's Hospital outside the submitted work. H.S. reports personal fees from AstraZeneca, Bayer Vital GmbH, Boehringer Ingelheim, Bristol Myers Squibb, Daiichi Sankyo, MSD, Novartis, Pharmacosmos, Sanofi, Servier, Synlab, Amgen and Amarin outside the submitted work. S.V.W. reports personal fees from MITRE, a federally funded research and development center for the Centers for Medicare and Medicaid Services and personal fees from Cytel Inc. during the conduct of the study. The other authors declare no competing interests.

## Additional information

**Correspondence and requests for materials** should be addressed to Nils Krüger.

# Reporting Summary

## Statistics

For all statistical analyses, confirm that the following items are present in the figure legend, table legend, main text, or Methods section.

| n/a | Confirmed | |
|---|---|---|
| ☐ | ☒ | The exact sample size (*n*) for each experimental group/condition, given as a discrete number and unit of measurement |
| ☐ | ☒ | A statement on whether measurements were taken from distinct samples or whether the same sample was measured repeatedly |
| ☒ | ☐ | The statistical test(s) used AND whether they are one- or two-sided *Only common tests should be described solely by name; describe more complex techniques in the Methods section.* |
| ☐ | ☒ | A description of all covariates tested |
| ☒ | ☐ | A description of any assumptions or corrections, such as tests of normality and adjustment for multiple comparisons |
| ☐ | ☒ | A full description of the statistical parameters including central tendency (e.g. means) or other basic estimates (e.g. regression coefficient) AND variation (e.g. standard deviation) or associated estimates of uncertainty (e.g. confidence intervals) |
| ☐ | ☒ | For null hypothesis testing, the test statistic (e.g. *F*, *t*, *r*) with confidence intervals, effect sizes, degrees of freedom and *P* value noted *Give P values as exact values whenever suitable.* |
| ☒ | ☐ | For Bayesian analysis, information on the choice of priors and Markov chain Monte Carlo settings |
| ☒ | ☐ | For hierarchical and complex designs, identification of the appropriate level for tests and full reporting of outcomes |
| ☒ | ☐ | Estimates of effect sizes (e.g. Cohen's *d*, Pearson's *r*), indicating how they were calculated |

*Our web collection on statistics for biologists contains articles on many of the points above.*

## Software and code

Policy information about availability of computer code

| Data collection | Routinely collected healthcare claims data was licensed via data use agreements for Medicare, Optum Clinformatics, and Merative MarketScan data |
|---|---|
| Data analysis | Aetion Evidence Platform, Python, R |

For manuscripts utilizing custom algorithms or software that are central to the research but not yet described in published literature, software must be made available to editors and reviewers. We strongly encourage code deposition in a community repository (e.g. GitHub). See the Nature Portfolio guidelines for submitting code & software for further information.

## Data

Policy information about availability of data

All manuscripts must include a data availability statement. This statement should provide the following information, where applicable:
- Accession codes, unique identifiers, or web links for publicly available datasets
- A description of any restrictions on data availability
- For clinical datasets or third party data, please ensure that the statement adheres to our policy

Data use agreements and licensing agreements do not allow sharing of patient-level claims data with third parties. However, data can be requested at the vendors directly (Optum Clinformatics, connected@optum.com; Medicare, resdac@umn.edu; Merative MarketScan, marketscan.support@merative.com).

# Research involving human participants, their data, or biological material

Policy information about studies with human participants or human data. See also policy information about sex, gender (identity/presentation), and sexual orientation and race, ethnicity and racism.

| | |
|---|---|
| Reporting on sex and gender | We report descriptive characteristics of our analytic cohort, including self-reported gender. Our table 1 describing baseline characteristics describes the count and percent of patients who are self-reported female or self-reported male. |
| Reporting on race, ethnicity, or other socially relevant groupings | We report descriptive characteristics such as self-reported race in our table 1. The self-reported race is included in our propensity score model and is balanced between exposure groups in the analytic cohort that produced the primary results. The self reported race categories include:<br>White; n (%)<br>Black; n (%)<br>Unknown / Missing; n (%)<br>Others; n (%) |
| Population characteristics | Patients with obesity and type 2 diabetes at elevated cardiovascular risk |
| Recruitment | Patients were identified from routinely collected healthcare claims data in a secondary data analysis. No patients were directly contacted or recruited. |
| Ethics oversight | The study was approved by the Mass General Brigham Institutional Review Board |

Note that full information on the approval of the study protocol must also be provided in the manuscript.

# Field-specific reporting

Please select the one below that is the best fit for your research. If you are not sure, read the appropriate sections before making your selection.

☒ Life sciences ☐ Behavioural & social sciences ☐ Ecological, evolutionary & environmental sciences

For a reference copy of the document with all sections, see nature.com/documents/nr-reporting-summary-flat.pdf

# Life sciences study design

All studies must disclose on these points even when the disclosure is negative.

| | |
|---|---|
| Sample size | Sample size and power calculations were included in the protocol that was pre-registered prior to conducting analyses. The assumptions of the power calculations are detailed in the protocols. |
| Data exclusions | Exclusions for the benchmarking studies emulated the eligibility criteria for the SUSTAIN-6 and SUPRASS-CVOT trials. Exclusions were relaxed for the expanded study populations. These criteria were pre-specified in the pre-registered protocols. |
| Replication | The benchmarking studies were attempts to replicate the SUSTAIN-6 and SURPASS-CVOT trials using non-randomized healthcare data. The trial results were replicated with the exception of the all-cause mortality outcome for the SUSTAIN-6 benchmarking emulation. This may have been related to potential residual confounding for this secondary end point that could reflect preferential prescribing in patients for whom clinicians anticipated limited life expectancy. Insights from this lack of replication informed the study in expanded populations and the head-to-head comparison. |
| Randomization | This was a non-randomized study that made secondary use of routinely collected healthcare claims data. |
| Blinding | There was no blinding in this non-randomized study that made secondary use of routinely collected healthcare claims data. |

# Reporting for specific materials, systems and methods

We require information from authors about some types of materials, experimental systems and methods used in many studies. Here, indicate whether each material, system or method listed is relevant to your study. If you are not sure if a list item applies to your research, read the appropriate section before selecting a response.

## Materials & experimental systems

| n/a | Involved in the study |
|---|---|
| ☒ | ☐ Antibodies |
| ☒ | ☐ Eukaryotic cell lines |
| ☒ | ☐ Palaeontology and archaeology |
| ☒ | ☐ Animals and other organisms |
| ☐ | ☒ Clinical data |
| ☒ | ☐ Dual use research of concern |
| ☒ | ☐ Plants |

## Methods

| n/a | Involved in the study |
|---|---|
| ☒ | ☐ ChIP-seq |
| ☒ | ☐ Flow cytometry |
| ☒ | ☐ MRI-based neuroimaging |

## Clinical data

Policy information about clinical studies

All manuscripts should comply with the ICMJE guidelines for publication of clinical research and a completed CONSORT checklist must be included with all submissions.

| | |
|---|---|
| Clinical trial registration | NCT06659744, NCT07088718, NCT07096063 |
| Study protocol | See ClinicalTrials.gov |
| Data collection | Three secondary healthcare claims databases, including data from Medicare Parts A, B, and D (2018 through 2020), Optum Clinformatics Data Mart (2018 through Feburary 2025), and Merative MarketScan (2018 through 2023). |
| Outcomes | The primary and secondary outcome measures were pre-specified in protocols that were pre-registered on clinicaltrials.gov. The end point algorithms were previously validated and showed a sensitivity of over 99% for mortality in the National Death Index, a positive predictive value of 94% for myocardial infarction, and 95% for stroke, and 98% for heart failure hospitalization. |

## Plants

| | |
|---|---|
| Seed stocks | No seed stocks were involved in this study. |
| Novel plant genotypes | No novel plant genotypes were involved in this study. |
| Authentication | No authentication of seed stocks or novel plant genotypes were relevant for this study. |

