## [Peer Review File · Nature Medicine]

Cardiovascular outcomes of semaglutide and tirzepatide for patients with type 2 diabetes in clinical practice

Corresponding Author: Dr Nils Krüger

Version 0:

Reviewer comments:

Reviewer #1

(Remarks to the Author)

A. Summary of key results: Emulation of SUSTAIN-6 and SURPASS-CVOT using U.S. claims data (5 cohorts) shows semaglutide and tirzepatide both reduce MACE versus comparators (sitagliptin as placebo proxy and dulaglutide, respectively), with similar benefits across the two drugs in head-to-head comparison (confidence intervals consistent with no difference, although point estimates indicating modest numeric advantage of semaglutide for reducing MACE especially among those at high CV risk, and point estimates for tirzepatide showed potential modest advantage for heart failure end points).

B. Originality and significance: Novel integration of trial emulation benchmarking, expanded population use, and real-world data to address evidence gaps (lack of head-to-head RCTs).

C. Data & methodology: Large datasets, prespecified protocols (NCT and Open Science Framework), transparency (GitHub code), frameworks (RCT-DUPLICATE, HARPER, RECORD-PE, STROBE), propensity matching, trial benchmarking strengthens validity.

D. Appropriate use of statistics and treatment of uncertainties: Analyses with Python, R, Aetion Evidence Platform. Potential residual confounding (e.g. SUSTAIN-6 disagreement for all-cause mortality) acknowledged. Claims-based limitations acknowledged. High treatment discontinuation rates ($\geq 34\%$) and relatively short median follow-up (median < 200 days for all drugs) may bias longer-term risk estimates.

E. Conclusions: Findings robust across most subgroups (except for tirzepatide versus dulaglutide estimates suggested male patients to benefit more from tirzepatide than female patients); mortality endpoints interpreted cautiously.

F. Suggested improvements (where possible):

- Extend follow-up (e.g. beyond > 200 days) to assess durability of benefit
- Examine treatment persistence sensitivity analysis
- Examine dosing effects
- Alternative comparator sensitivity analysis (other than sitagliptin as placebo proxy)
- Effect of drop-in treatments over treatment period (e.g. SGLT2i)
- Stratify heart failure outcomes by phenotype (e.g. preserved versus reduced ejection fraction)
- Consider mention limitations in generalisability of these data outwith U.S. to international populations
- Carefully check for typos e.g. Figure 2 title "expaned" should be "expanded"

G. References: Appear comprehensive and appropriate.

H. Clarity and context: Clear and comprehensive.

(Remarks on code availability)

GitHub website specifically says "The code provided here is not sufficient to replicate study results, as it: Does not include source data (privacy and data-use restrictions)"

Reviewer #2

(Remarks to the Author)

Major points

- 1) Tirzepatide and semaglutide are prescribed for different reasons to different patient cohorts. Propensity-score matching cannot account for these differences.
- 2) Sitagliptin is not a good choice of placebo proxy. People are prescribed DPP-IV inhibitors for very different reasons.
- 3) We need prospective trials in broad populations. The limitations of this analysis are very large.
- 4) The benchmarking exercise did not provide close approximations to the original trial results. The hazard ratios were not close. This is despite what is stated in the paper.
- 5) SURPASS-CVOT results are not yet available so there are "pending results".
- 6) The team recognize that data from administrative claims are not as accurate as captured in trials. Have they validated the various data points in the trials? The endpoint validation is given for some but not endpoints. They are not very impressive. A notably absent endpoint re validation is heart failure event.
- 7) To validate such an approach the team could have attempted to reproduce many other CV outcome trials. There are of course many with GLP-1 RAs as well as many other interventions.

(Remarks on code availability)

XX

Reviewer #3

(Remarks to the Author)

Summary of the key results: cohort studies can be conducted to emulate clinical trials and try to overcome their limitations; semaglutide exerts greater CV protection vs. sitagliptin, tirzepatide exerts similar CV effects compared to dulaglutide and semaglutide

Originality and significance: using real-world data to perform trial emulation studies is relevant but not entirely new (DOI: 10.7326/ANNALS-24-00775)

Data & methodology: the major concern here regards follow-up duration. Mean follow-up on intervention drug is approximately 6 months, which is the time needed to reach the maximum dose of tirzepatide. Hence we are likely comparing MTD of semaglutide and Lower doses of tirzepatide (unlike what happened in the SURPASS CVOT trial, where patients were titrated up to 15 mg and most of them stayed on that dose). Also, when evaluating CV outcomes with GLP-1RA, one year of follow-up is definitely too short of a timeframe: if you look at Kaplan Meier curves in REWIND you will see that the curves only start to diverge at 1 year and their distance increases overtime. All in all, this design does not allow a fair comparison between tirzepatide and GLP-1RA.

Appropriate use of statistics and treatment of uncertainties: Management of missing data should be described more accurately, I could not find any reference to it in the manuscript.

Conclusions: speculations on the fact that tirzepatide may be less CV protective as exhibiting a lower affinity for GLP-1RA are not accurate. It is true that tirzepatide has lower affinity for GLP-1R than native GLP-1, but it is also characterized by biased agonism meaning that it does not activate the beta-arresting pathway, hence inhibiting GLP-1R internalization and prolonging and enhancing its favorable metabolic effects. GIPR agonism has also been proven as anti-atherosclerotic.

Suggested improvements: extending follow-up time is mandatory as far as I am concerned

References: appropriate.

(Remarks on code availability)

I do not have the expertise to evaluate the code.

Reviewer #4

(Remarks to the Author)

I appreciate the opportunity to review this reference trial emulation, benchmarking and extension/expansion study on the cardiovascular effects of Semaglutide and Tirzepatide, showing that semaglutide (vs sitagliptin) reduced MACE and tirzepatide was comparable to dulaglutide at reducing MACE in patients at elevated CV risk, and comparable effects of tirzepatide versus semaglutide.

The study is novel and uses appropriate data sources and methodology to address the research questions. The conclusions made are valid.

There are a number of generally minor points with the study that I would expect to see updated before publication, which are detailed below:

Abstract and Introduction

1. In the Abstract, please add detail on what the "expanded populations routinely seen in clinical practice" were for to help provide a clear overview of who was studied in the analyses. Same applies to Introduction paragraph three.

Methods

2. Data sources - please include a high-level description (1 paragraph max for each) of each of the reference trials under the heading "The reference trials", in order to orientate the reader early on in relation to the characteristics, high-level results and

the rationale for selection of the reference trials.

3. Paragraph three of the "Specification and emulation of the studies" section states that the TARGET guideline was followed. In TARGET, the target trial is specifically defined as a "hypothetical pragmatic randomized trial". In this study SUSTAIN-6 and SURPASS-CVOT are not hypothetical trials but actual trials. Throughout this paper, the term reference trial has been used which seems appropriate and consistent with other studies in the literature - please therefore update table 3 so that the term "reference trial" is used instead of "target trial" to clarify that emulation of SUSTAIN-6 is emulation of an actual trial (so existing headings would become "reference trial specification" and "reference trial emulation in US claims data"). In keeping with this and with TARGET, the expanded population analysis would be considered emulation of a (hypothetical) target trial, and in Table 3 this should also be indicated by including a column 3 and 4 ("Expanded population target trial" and "Expanded population target trial emulation in US claims data") which the current "Expanded population" content can be placed in.

4. Table 3 contains the specification of one of the three protocols mentioned in paragraph. Please include the other two in the appendix.

Results and Discussion

5. In the "Benchmarking against SUSTAIN-6 and SURPASS-CVOT" section, it is stated "divergent results for all-cause mortality suggestive of residual confounding". This is discussed again in paragraph four of the discussion. "Residual confounding" is quite a general, non-specific explanation. Could the authors provide a little more detail on what the confounder(s) may be here? I guess an alternative possibility could be that the placebo-proxy sitagliptin is not a good placebo when considering death as an outcome, and that in fact placebo-proxy comparators are non-ideal when measuring death as an outcome as all-cause mortality is actually a non-specific outcome? This could be investigated and then commented on by performing a post-hoc analysis assessing causes of death in each of the semaglutide and sitagliptin groups (or this point at least discussed if these data are not available).

(Remarks on code availability)

Version 1:

Reviewer comments:

Reviewer #1

(Remarks to the Author)

The reviewer comments have all been comprehensively and adequately addressed, and the manuscript is now much improved. I have no further comments.

(Remarks on code availability)

Reviewer #4

(Remarks to the Author)

All comments have been well addressed.

(Remarks on code availability)

Reviewers' Comments:

Reviewer #1 (Remarks to the Author):

A. Summary of key results: Emulation of SUSTAIN-6 and SURPASS-CVOT using U.S. claims data (5 cohorts) shows semaglutide and tirzepatide both reduce MACE versus comparators (sitagliptin as placebo proxy and dulaglutide, respectively), with similar benefits across the two drugs in head-to-head comparison (confidence intervals consistent with no difference, although point estimates indicating modest numeric advantage of semaglutide for reducing MACE especially among those at high CV risk, and point estimates for tirzepatide showed potential modest advantage for heart failure end points).

B. Originality and significance: Novel integration of trial emulation benchmarking, expanded population use, and real-world data to address evidence gaps (lack of head-to-head RCTs).

C. Data & methodology: Large datasets, prespecified protocols (NCT and Open Science Framework), transparency (GitHub code), frameworks (RCT-DUPLICATE, HARPER, RECORD-PE, STROBE), propensity matching, trial benchmarking strengthens validity.

D. Appropriate use of statistics and treatment of uncertainties: Analyses with Python, R, Aetion Evidence Platform. Potential residual confounding (e.g. SUSTAIN-6 disagreement for all-cause mortality) acknowledged. Claims-based limitations acknowledged. High treatment discontinuation rates ($\geq 34\%$) and relatively short median follow-up (median < 200 days for all drugs) may bias longer-term risk estimates.

E. Conclusions: Findings robust across most subgroups (except for tirzepatide versus dulaglutide estimates suggested male patients to benefit more from tirzepatide than female patients); mortality endpoints interpreted cautiously.

F. Suggested improvements (where possible):

Thank you for the suggested analyses. We have added additional analyses where possible.

- Extend follow-up (e.g. beyond > 200 days) to assess durability of benefit

We prespecified the primary end point analysis around an on-treatment causal contrast, where follow-up would end at the time of treatment discontinuation (plus a 45-day grace and risk window) due to our experience and published literature (1, 2) that patients in clinical practice have poor adherence and short time on-treatment for GLP-1-based medication.

As a sensitivity analysis, we have also prespecified an analysis that targets the as-started causal contrast where we follow the patients for a fixed time of 1 year, unless an outcome event occurs. Results of this analysis were part of the initial submission and show results that are very much in line with the main analysis with estimates that are slightly attenuated, as one would expect for an analog of an intention-to-treat analysis.

Following your suggesting, we have now also included an expanded 2 year on-treatment sensitivity analysis that follows the patients, who do stay on treatment for a prolonged time. Results of this analysis are as expected very concordant with the 1-year on-treatment analysis due to the high attrition of patients who become non-adherent within the first year. We have accordingly documented this additional analysis as an amendments to the study protocols and have registered these in ClinicalTrials.gov

Primary outcome (pooled estimates)

Study cohort	1-year on-treatment primary analysis, HR with 95% CI	2-year on-treatment sensitivity analysis, HR with 95% CI
SUSTAIN-6 emulation	0.68 (0.60 to 0.77)	0.65 (0.59 to 0.72)
SUSTAIN-6 expanded population	0.82 (0.74 to 0.91)	0.81 (0.73 to 0.89)
SURPASS-CVOT emulation	0.83 (0.69 to 1.01)	0.79 (0.66 to 0.94)
SURPASS-CVOT expanded population	0.87 (0.75 to 1.01)	0.84 (0.73 to 0.96)
TIRZSEMA-CVOT	1.06 (0.95 to 1.18)	1.04 0.94 to 1.16)

- (1) Hankosky ER, Chinthammit C, Meeks A, et al. Real-world use and effectiveness of tirzepatide among individuals without type 2 diabetes: Results from the Optum Market Clarity database. *Diabetes, Obesity and Metabolism*. 2025;27(5):2810-2821. doi:[10.1111/dom.16290](https://doi.org/10.1111/dom.16290)
- (2) Ladebo L, Ernst MT, Mailhac A, Dirksen C, Bojsen-Møller KN, Pottegård A. Real-World Use of Semaglutide for Weight Management: Patient Characteristics and Dose Titration—A Danish Cohort Study. *Diabetes Care*. 2024;47(10):1834-1837. doi:[10.2337/dc24-1082](https://doi.org/10.2337/dc24-1082)

- Examine treatment persistence sensitivity analysis

Thank you. We believe we have already addressed this by reporting mean (SD) and median (IQR) follow-up time for our analysis. While treatment persistence was short, this is reflective of clinical practice patterns of the US population.

From the manuscript, paragraph of the primary end point in expanded populations:
Sema vs sita: mean follow-up on-treatment for semaglutide users was 193 days (median = 157 days; interquartile range [IQR] = 85 to 331 days) and for sitagliptin users 195 days (median = 160; 95 to 322). Treatment discontinuation (46%) was the most common reason for censoring.

Tirz vs dula: Among tirzepatide users, the pooled median on-treatment follow-up was 189 days (median = 162; 72 to 321). For dulaglutide users, the corresponding follow-up was on average 173 days (median = 139; 73 to 257). Discontinuation of treatment (37%) was the most common reason for censoring.

Tirz vs sema: The mean follow-up on-treatment was 181 days (median = 155; 71 to 290) for tirzepatide initiators and 174 days (108) (median = 148; 82 to 254) for semaglutide initiators. Treatment discontinuation (34%) was the most common censoring reason.

- Examine dosing effects

This study focuses on the use of injectable semaglutide, tirzepatide, dulaglutide, and sitagliptin in clinical practice. Therefore, our study does not examine dose-specific effects for the following reasons: (1) The research question being asked in this analysis is whether initiation of either of these medications reduces the risk of major adverse cardiovascular outcomes in routine care (as indicated in the title), regardless of up- or down-titration after time zero. Considering time-dependent titration would reflect a different research question that would require other analytic approaches that were not the focus of this analysis. (2) Dosage information in pharmacy claims reflects dispensed products rather than the actual titrated dose taken, and dose escalation patterns for tirzepatide and semaglutide in clinical practice are highly heterogeneous and often incomplete. As shown in other observational studies, only a minority of patients reach the maximum maintenance doses used in trials, with approximately one-third remaining on non-maximum dosages in both semaglutide as well as tirzepatide after several months of therapy. This under-titration occurs for clinical, tolerability, and supply reasons, resulting in overlapping exposure ranges that preclude clear dose stratification. (3) Evidence from randomized trials demonstrates that both agents exert clinically meaningful effects well before maximum doses are achieved with only about 2/3 reaching maximum dosages in RCTs. Given the absence of reliable longitudinal titration data and the lack of a validated framework for defining “equivalent” doses across the two drug classes, any attempt to compare dose-response effects would be prone to misclassification and bias. Therefore, our analysis focuses on treatment initiators under routine clinical use, which best reflects effectiveness in routine practice.

- Alternative comparator sensitivity analysis (other than sitagliptin as placebo proxy)

Thank you for this suggestion. Sitagliptin, a dipeptidyl peptidase-4 (DPP-4) inhibitor, was selected as an active-comparator placebo proxy based on extensive validation in prior work.

This approach aligns with insights from the RCT-DUPLICATE initiative and other observational studies that found sitagliptin to be a good placebo proxy in the setting of CV outcomes. Importantly, sitagliptin has been shown to have a neutral effect on major cardiovascular and heart failure outcomes in the TECOS trial. Sitagliptin also distinguishes itself from other agents within the same class (e.g., saxagliptin and linagliptin) that may introduce indication bias. Lastly, benchmarking in the studies presented here suggest that sitagliptin is a valid proxy for placebo in this setting. We agree that exploring additional comparators may be of future interest and appreciate the reviewer's insightful comment.

- Effect of drop-in treatments over treatment period (e.g. SGLT2i)

Thank you for the thoughtful suggestion. Importantly, our analyses were designed emulate the design of the SUSTAIN-6 and SURPASS-CVOT trials. Both the SUSTAIN-6 protocol (1) and the SURPASS-CVOT design and process paper (2) specify that study drugs were administered on top of standard-of-care therapy, allowing site investigators to initiate or adjust concomitant glucose-lowering agents, including SGLT2 inhibitors, as clinically indicated. Our emulation mirrors this pragmatic design feature. Therefore, drop-in of comedication during follow-up were not the focus of this investigation. Consistent with the randomization procedure of the trials, we balanced on baseline medication use, including SGLT2 inhibitors. We agree that post-baseline comedication patterns are of scientific interest, and this valuable comment may inform future research.

- (1) Marso SP, Bain SC, Consoli A, et al. Semaglutide and Cardiovascular Outcomes in Patients with Type 2 Diabetes. Study protocol. *N Engl J Med*. 2016;375(19):1834-1844. doi:[10.1056/NEJMoa1607141](https://doi.org/10.1056/NEJMoa1607141)
- (2) Nicholls SJ, Bhatt DL, Buse JB, et al. Comparison of tirzepatide and dulaglutide on major adverse cardiovascular events in participants with type 2 diabetes and atherosclerotic cardiovascular disease: SURPASS-CVOT design and baseline characteristics. *American Heart Journal*. 2024;267:1-11. doi:[10.1016/j.ahj.2023.09.007](https://doi.org/10.1016/j.ahj.2023.09.007)

We have also added this information into Table 3 (and Table 9 and 10 in the Supplementary Information) where we specified the (target) trials and their emulations

“Glucose-lowering therapy for each participant was allowed to be modified at the discretion of the site investigator through open-label initiation or adjustment of non-GLP-1 receptor agonist medications, with individualized HbA1c targets set according to local clinical practice and current professional society guidelines.”

- Stratify heart failure outcomes by phenotype (e.g. preserved versus reduced ejection fraction)

The focus of our analysis was major adverse cardiovascular events (MACE), consistent with the end points evaluated in the trials we emulated. Hospitalization for heart failure was

examined as a secondary outcome and is reported accordingly. Stratifying heart failure outcomes by phenotype (e.g., preserved versus reduced ejection fraction) is certainly interesting; however, detailed ejection fraction data are not available in the data sources used for this study, and such analyses were beyond the scope of the present work. We agree that this represents an important area for future investigation and appreciate the reviewer's insightful comment.

- Consider mention limitations in generalizability of these data outwith U.S. to international populations

Thank you for this suggestion. We added a sentence to the limitations section:

"Fourth, although our data represent a diverse population, findings may best apply to the United States and have more limited applicability to international settings."

- Carefully check for typos e.g. Figure 2 title "expaned" should be "expanded"

Thank you, we updated the figure accordingly and checked for other typos.

G. References: Appear comprehensive and appropriate.

H. Clarity and context: Clear and comprehensive.

Reviewer #1 (Remarks on code availability):

GitHub website specifically says "The code provided here is not sufficient to replicate study results, as it: Does not include source data (privacy and data-use restrictions)"

Reviewer #2 (Remarks to the Author):

Major points

1) Tirzepatide and semaglutide are prescribed for different reasons to different patient cohorts. Propensity-score matching cannot account for these differences.

Thank you for this comment. While differences in prescribing patterns may exist, the clinical indications for tirzepatide and semaglutide are largely similar. In our data, we observed only modest differences in measured baseline characteristics between treatment groups before

matching and almost none after propensity-score matching. This demonstrates that our matching procedure effectively balanced observed covariates, addressing population differences to the extent possible in an observational setting. Although residual confounding from unmeasured factors cannot be entirely ruled out, the high degree of concordance across all stages of our trial emulation provides confidence in the robustness of our findings.

2) Sitagliptin is not a good choice of placebo proxy. People are prescribed DPP-IV inhibitors for very different reasons.

Thank you for this suggestion. Sitagliptin, a dipeptidyl peptidase-4 (DPP-4) inhibitor, was selected as an active-comparator placebo proxy based on extensive validation in prior work. This approach aligns with insights from the RCT-DUPLICATE initiative and other observational studies that found sitagliptin to be a good placebo proxy in the setting of CV outcomes. Importantly, sitagliptin has been shown to have a neutral effect on major cardiovascular and heart failure outcomes in the TECOS trial. Sitagliptin also distinguishes itself from other agents within the same class (e.g., saxagliptin and linagliptin) that may introduce indication bias. Lastly, benchmarking in the studies presented here suggest that sitagliptin is a valid proxy for placebo in this setting. We agree that exploring additional comparators may be of future interest and appreciate the reviewer's insightful comment.

3) We need prospective trials in broad populations. The limitations of this analysis are very large.

We agree that large, prospective trials in broad populations are essential. In the meantime, clinicians still need evidence to guide decisions. Our study addresses this gap by emulating the trial protocols (new-user, active-comparator design; rigorous propensity-score balancing; validated cardiovascular endpoints) and by benchmarking against corresponding RCTs, where our estimates were concordant.

4) The benchmarking exercise did not provide close approximations to the original trial results. The hazard ratios were not close. This is despite what is stated in the paper.

To assess concordance between our database emulations and their randomized trial counterparts, we compared the primary endpoint results with the published findings from SUSTAIN-6 and, once results became available, from SURPASS-CVOT. Agreement between the trial and real-world evidence estimates was evaluated using three prespecified binary metrics defined by the RCT-DUPLICATE initiative:

1. Statistical agreement, defined as both the database study and trial estimates and their 95% confidence intervals lying on the same side of the null (assessed for primary end points only, secondary end points were not powered for statistical agreement);
2. Directional agreement, defined as both the database study and trial estimates lying on the same side of the null;

3. Estimate agreement, defined as the estimate of the database study falling within the 95% confidence interval of the trial;
3. Standardized difference agreement, defined as an absolute standardized difference $|Z| < 1.96$, with $Z = \frac{\hat{\theta}_{RCT} - \hat{\theta}_{RWE}}{\sqrt{\hat{\sigma}_{RCT}^2 + \hat{\sigma}_{RWE}^2}}$ where $\hat{\theta}_{RCT}$ and $\hat{\theta}_{RWE}$ are the treatment effect estimates with associated variances $\hat{\sigma}^2$.

The concordance for the majority of the endpoints in benchmarking and discordance for all-cause mortality in the SUSTAIN-6 emulation were used to inform our subsequent analyses.

	End point	Database estimate	Trial estimate	SA	DA	EA	SD
SUSTAIN-6	Major adverse cardiovascular events	0.68 (0.60 to 0.77)	0.74 (0.58 to 0.95)	Yes	Yes	Yes	Yes
	Myocardial infarction	0.70 (0.57 to 0.86)	0.74 (0.51–1.08)	N/A*	Yes	Yes	Yes
	Stroke	0.82 (0.65 to 1.03)	0.61 (0.38–0.99)	N/A*	Yes	Yes	Yes
	All-cause mortality	0.58 (0.48 to 0.71)	1.05 (0.74 to 1.50)	N/A*	No	No	No
SURPASS-CVOT	Major adverse cardiovascular events	0.83 (0.69 to 1.01)	0.92 (0.83 to 1.01)	Yes	Yes	Yes	Yes
	Myocardial infarction	0.81 (0.61 to 1.06)	0.86 (0.74 to 1.00)	N/A*	Yes	Yes	Yes
	Stroke	0.92 (0.65 to 1.29)	0.91 (0.76 to 1.09)	N/A*	Yes	Yes	Yes
	All-cause mortality	0.76 (0.52 to 1.11)	0.84 (0.75 to 0.94)	N/A*	Yes	Yes	Yes

5) SURPASS-CVOT results are not yet available so there are “pending results”.

The remaining results for the components of the composite end point became available in the meantime and show highly concordant results. We have accordingly updated the table in the manuscript.

6) The team recognize that data from administrative claims are not as accurate as captured in trials. Have they validated the various data points in the trials? The endpoint validation is given for some but

not endpoints. They are not very impressive. A notably absent endpoint re validation is heart failure event.

We agree that outcome ascertainment in administrative data requires careful validation. All endpoints used in this analysis were based on previously validated claims-based algorithms with high positive predictive value. Specifically, our endpoint definitions demonstrated a sensitivity of over 99% for mortality (validated against the National Death Index), a positive predictive value of 94% for myocardial infarction, 95% for stroke, and 98% for heart failure hospitalization.

7) To validate such an approach the team could have attempted to reproduce many other CV outcome trials. There are of course many with GLP-1 RAs as well as many other interventions.

Thank you for this suggestion. We agree that reproducing results from multiple cardiovascular outcome trials is an important validation step for any emulation framework. We did exactly that in the RCT-DUPLICATE initiative, in which we emulated a broad range of randomized clinical trials across cardiovascular and metabolic diseases—including several involving GLP-1 receptor agonists and the same cardiovascular outcomes assessed here. We therefore feel confident that the current design, data, and analytics set up rest on a well-validated foundation.

Wang SV, Schneeweiss S, RCT-DUPLICATE Initiative, et al. Emulation of Randomized Clinical Trials With Nonrandomized Database Analyses: Results of 32 Clinical Trials. *JAMA*. 2023;329(16):1376-1385. doi:[10.1001/jama.2023.4221](https://doi.org/10.1001/jama.2023.4221)

Krüger N, Schneeweiss S, Fuse K, et al. Semaglutide and Tirzepatide in Patients With Heart Failure With Preserved Ejection Fraction. *JAMA*. Published online August 31, 2025. doi:[10.1001/jama.2025.14092](https://doi.org/10.1001/jama.2025.14092)

Reviewer #2 (Remarks on code availability):

XX

Reviewer #3 (Remarks to the Author):

Summary of the key results: cohort studies can be conducted to emulate clinical trials and try to overcome their limitations; semaglutide exerts greater CV protection vs. sitagliptin, tirzepatide exerts similar CV effects compared to dulaglutide and semaglutide

Originality and significance: using real-world data to perform trial emulation studies is relevant but not entirely new (DOI: 10.7326/ANNALS-24-00775)

Thank you for this comment. We agree that target trial emulations have become an increasingly important approach, as reflected in recent studies such as Saeed *et al.* (Ann Intern Med 2025; DOI: 10.7326/ANNALS-24-00775), which compared SGLT2 inhibitors and GLP-1 receptor agonists for cardiovascular outcomes.

Our work advances the field in several important ways. First, we directly emulated, predicted, and benchmarked against two cardiovascular outcome trials—SUSTAIN-6 and SURPASS-CVOT—using a rigorously prespecified framework, enabling transparent evaluation of concordance with reference RCTs before expanding analyses to broader clinical populations. Second, we provide a head-to-head comparison of tirzepatide versus semaglutide for major adverse cardiovascular events, a question not yet addressed by randomized trials. Finally, by embedding trial benchmarking and expanded analyses within a single methodological framework, our study demonstrates a reproducible approach for integrating real-world evidence into cardiovascular drug evaluation. Together, we believe these features make this work a timely and significant contribution.

Data & methodology: the major concern here regards follow-up duration. Mean follow-up on intervention drug is approximately 6 months, which is the time needed to reach the maximum dose of tirzepatide. Hence we are likely comparing MTD of semaglutide and Lower doses of tirzepatide (unlike what happened in the SURPASS CVOT trial, where patients were titrated up to 15 mg and most of them stayed on that dose). Also, when evaluating CV outcomes with GLP-1RA, one year of follow-up is definitely too short of a timeframe: if you look at Kaplan Meier curves in REWIND you will see that the curves only start to diverge at 1 year and their distance increases overtime. All in all, this design does not allow a fair comparison between tirzepatide and GLP-1RA.

Thank you for this thoughtful comment. In clinical practice, patients rarely reach or remain on the maximum approved doses of GLP-1–based medications, including both semaglutide and tirzepatide, as shown in multiple studies (1, 2). Therefore, our primary analysis was prespecified around an on-treatment causal contrast, with follow-up ending at treatment discontinuation (plus a 45-day grace and risk window), to reflect medication use patterns from clinical practice. As a sensitivity analysis, we also prespecified an as-started causal contrast with fixed 1-year follow-up, analogous to an intention-to-treat analysis. Results from this approach were highly consistent with the main analysis, showing only the expected mild attenuation of effect estimates.

Following your suggestion, we have now additionally conducted an extended 2-year on-treatment sensitivity analysis among patients who remained on therapy for a prolonged period. Results were very similar to the 1-year analysis, indicating that extending follow-up among treatment-persistent users does not meaningfully alter the observed comparative cardiovascular risk estimates. We have accordingly documented this additional analysis in the study protocols and have registered the amendments in ClinicalTrials.gov

Primary outcome (pooled estimates)

Study cohort	1-year on-treatment primary analysis, HR with 95% CI	2-year on-treatment sensitivity analysis, HR with 95% CI
SUSTAIN-6 emulation	0.68 (0.60 to 0.77)	0.65 (0.59 to 0.72)
SUSTAIN-6 expanded population	0.82 (0.74 to 0.91)	0.81 (0.73 to 0.89)
SURPASS-CVOT emulation	0.83 (0.69 to 1.01)	0.79 (0.66 to 0.94)
SURPASS-CVOT expanded population	0.87 (0.75 to 1.01)	0.84 (0.73 to 0.96)
TIRZSEMA-CVOT	1.06 (0.95 to 1.18)	1.04 (0.94 to 1.16)

- (1) Hankosky ER, Chinthammit C, Meeks A, et al. Real-world use and effectiveness of tirzepatide among individuals without type 2 diabetes: Results from the Optum Market Clarity database. *Diabetes, Obesity and Metabolism*. 2025;27(5):2810-2821. doi:[10.1111/dom.16290](https://doi.org/10.1111/dom.16290)
- (2) Ladebo L, Ernst MT, Mailhac A, Dirksen C, Bojsen-Møller KN, Pottegård A. Real-World Use of Semaglutide for Weight Management: Patient Characteristics and Dose Titration—A Danish Cohort Study. *Diabetes Care*. 2024;47(10):1834-1837. doi:[10.2337/dc24-1082](https://doi.org/10.2337/dc24-1082)

Appropriate use of statistics and treatment of uncertainties: Management of missing data should be described more accurately, I could not find any reference to it in the manuscript.

In claims, missing data are usually handled by assuming absence of a code indicated absence of the condition. Missing indicators were included for race and region in the propensity score model.

Conclusions: speculations on the fact that tirzepatide may be less CV protective as exhibiting a lower affinity for GLP-1RA are not accurate. It is true that tirzepatide has lower affinity for GLP-1R than native GLP-1, but it is also characterized by biased agonism meaning that it does not activate the beta-arresting pathway, hence inhibiting GLP-1R internalization and prolonging and enhancing its favorable metabolic effects. GIPR agonism has also been proven as anti-atherosclerotic.

Thank you for this comment. We agree that the precise mechanistic link between receptor affinity, biased agonism, and cardiovascular protection remains uncertain. Our intent was to acknowledge that tirzepatide and semaglutide differ in their GLP-1 receptor binding and signaling properties. We have therefore refined the language in the revised manuscript to reflect this uncertainty and maintain neutrality

“Tirzepatide binds the GLP-1 receptor with lower affinity than semaglutide and exhibits distinct signaling bias, which may result in comparatively different downstream GLP-1 receptor signaling dynamics.”

Suggested improvements: extending follow-up time is mandatory as far as I am concerned

Please see our comment from above. We have now included an extended follow-up analysis.

References: appropriate.

Reviewer #3 (Remarks on code availability):

I do not have the expertise to evaluate the code.

Reviewer #4 (Remarks to the Author):

I appreciate the opportunity to review this reference trial emulation, benchmarking and extension/expansion study on the cardiovascular effects of Semaglutide and Tirzepatide, showing that semaglutide (vs sitagliptin) reduced MACE and tirzepatide was comparable to dulaglutide at reducing MACE in patients at elevated CV risk, and comparable effects of tirzepatide versus semaglutide.

The study is novel and uses appropriate data sources and methodology to address the research questions. The conclusions made are valid.

There are a number of generally minor points with the study that I would expect to see updated before publication, which are detailed below:

Abstract and Introduction

1. In the Abstract, please add detail on what the “expanded populations routinely seen in clinical practice” were for to help provide a clear overview of who was studied in the analyses. Same applies to Introduction paragraph three.

Thank you for this suggestion. Due to word constraints in the Abstract, we are unable to include much detail on the expanded populations. However, we rephrased the sentence to more clearly convey their clinical relevance: *“Second, we assessed each drug in expanded populations reflective of patients routinely seen in clinical practice.”*

Methods

2. Data sources - please include a high-level description (1 paragraph max for each) of each of the reference trials under the heading "The reference trials", in order to orientate the reader early on in relation to the characteristics, high-level results and the rationale for selection of the reference trials.

Thank you for this suggestion. We added such a paragraph in the “Specification and emulation of the studies” paragraph.

“SUSTAIN-6 evaluated once-weekly semaglutide versus placebo in patients at moderate and high cardiovascular risk with type 2 diabetes, showing a 26% relative risk reduction in major adverse cardiovascular events (HR 0.74; 95% CI 0.58–0.95). SURPASS-CVOT compared once-weekly tirzepatide with dulaglutide in adults with type 2 diabetes and high cardiovascular risk, showing a 9% relative risk reduction in major adverse cardiovascular events (HR 0.92; 95% CI 0.83–1.01).”

3. Paragraph three of the “Specification and emulation of the studies” section states that the TARGET guideline was followed. In TARGET, the target trial is specifically defined as a "hypothetical pragmatic randomized trial". In this study SUSTAIN-6 and SURPASS-CVOT are not hypothetical trials but actual trials. Throughout this paper, the term reference trial has been used which seems appropriate and consistent with other studies in the literature - please therefore update table 3 so that the term “reference trial” is used instead of “target trial” to clarify that emulation of SUSTAIN-6 is emulation of an actual trial (so existing headings would become “reference trial specification” and “reference trial emulation in US claims data”). In keeping with this and with TARGET, the expanded population analysis would be considered emulation of a (hypothetical) target trial, and in Table 3 this should also be indicated by including a column 3 and 4 (“Expanded population target trial” and “Expanded population target trial emulation in US claims data”) which the current “Expanded population” content can be placed in.

We like this idea. Thank you very much. Necessary changes have been made.

4. Table 3 contains the specification of one of the three protocols mentioned in paragraph. Please include the other two in the appendix.

We have added two supplementary tables.

Results and Discussion

5. In the “Benchmarking against SUSTAIN-6 and SURPASS-CVOT” section, it is stated “divergent results for all-cause mortality suggestive of residual confounding”. This is discussed again in paragraph four of the discussion. “Residual confounding” is quite a general, non-specific explanation. Could the authors provide a little more detail on what the confounder(s) may be here? I guess an alternative possibility could be that the placebo-proxy sitagliptin is not a good placebo when considering death as an outcome, and that in fact placebo-proxy comparators are non-ideal when measuring death as an outcome as all-cause mortality is actually a non-specific outcome? This could be investigated and then commented on by performing a post-hoc analysis assessing causes of death in each of the semaglutide and sitagliptin groups (or this point at least discussed if these data are not available).

Thank you for this valuable comment. It is very difficult to disentangle the divergence in all-cause mortality between our SUSTAIN-6 emulation and the original trial. While multiple factors could contribute, including the non-specific nature of all-cause mortality and the limitations of using sitagliptin as a placebo proxy, we have added more context with one plausible explanation. Specifically, we now note that the difference may reflect preferential prescribing in patients for whom clinicians anticipated limited life expectancy:

“However, benchmarking against SUSTAIN-6 flagged disagreement for the effect on all-cause mortality, highlighting potential residual confounding for this secondary end point that could reflect preferential prescribing in patients for whom clinicians anticipated limited life expectancy.”

Editors' comments:

1. In line with comments from referees #1 and #3, we ask that you strongly consider providing additional data in which the follow-up time of treatment is extended, as stated by referee #1, section F.

Following the collective suggestions, we have included an expended 2 year on-treatment sensitivity analysis that follows the patients, who do stay on treatment for a prolonged time. Results of this analysis are as expected very concordant with the 1-year on-treatment analysis. We have accordingly documented this additional analysis in the study protocols and have registered the amendments in ClinicalTrials.gov

Primary outcome (pooled estimates)

Study cohort	1-year on-treatment primary analysis, HR with 95% CI	2-year on-treatment sensitivity analysis, HR with 95% CI
SUSTAIN-6 emulation	0.68 (0.60 to 0.77)	0.65 (0.59 to 0.72)
SUSTAIN-6 expanded population	0.82 (0.74 to 0.91)	0.81 (0.73 to 0.89)
SURPASS-CVOT emulation	0.83 (0.69 to 1.01)	0.79 (0.66 to 0.94)
SURPASS-CVOT expanded population	0.87 (0.75 to 1.01)	0.84 (0.73 to 0.96)
TIRZSEMA-CVOT	1.06 (0.95 to 1.18)	1.05 0.94 to 1.16)

2. Please provide some discussion of your recent JAMA paper (ref. 25).

The second discussion paragraph was adjusted to respond to this valuable suggestion:

“Conversely, point estimates for tirzepatide showed a potential modest advantage for heart failure end points, with confidence intervals compatible with no difference, consistent with recent data supporting protective effects of semaglutide and tirzepatide on heart failure outcomes and a potential incremental benefit with the latter.²⁵ Potential explanations include the dual receptor agonism of tirzepatide.²²”

3. Table 3 provides information on the "Specification and emulation of the SUSTAIN-6 trial including our expansion study". However, we did not see a similar table for the emulation and expansion study regarding the SURPASS-CVOT trial, and we suggest that this is provided.

We have added the two requested tables in the supplement.

4. On page 6, it is stated that "Among tirzepatide users, the pooled median on-treatment follow-up was 189 days (median = 162; 72 to 321)." Should the first instance of the word "median" be changed to "mean"?

Thank you, this is correct. We updated it to “mean” instead of “median”.

5. The Abstract should be formatted as a single paragraph, generally 200 words or shorter in length, with 250 words as a hard limit.

We updated the abstract to be within that limit.

6. We allow for a maximum of 6 main display items (figures + tables). As there are currently 4 figures and 3 tables, please reorganize these to reduce the overall number (for example, Table 2 could be changed to an Extended Data table).

We have followed your recommendation and moved the suggested table into the Supplementary Information (now this table is Extended Data Table 8).

7. The "Appendix" should be labeled as "Supplementary Information" and the figures in this file should be labeled as Extended Data Figures.

We have updated the nomenclature accordingly.

8. Please note that we allow for a maximum of 10 Extended Data Figures + Tables. Additional figures and tables should be included as Supplementary Figures and Tables.

We have updated the nomenclature accordingly